# Polylogarithmic width suffices for gradient descent to achieve arbitrarily small test error with shallow ReLU networks

**Ziwei Ji & Matus Telgarsky**
University of Illinois, Urbana-Champaign
{ziweiji2,mjt}@illinois.edu

## Abstract

Recent theoretical work has guaranteed that overparameterized networks trained by gradient descent achieve arbitrarily low training error, and sometimes even low test error. The required width, however, is always polynomial in at least one of the sample size $n$, the (inverse) target error $1/\epsilon$, and the (inverse) failure probability $1/\delta$. This work shows that $\widetilde{\Theta}(1/\epsilon)$ iterations of gradient descent with $\widetilde{\Omega}(1/\epsilon^2)$ training examples on two-layer ReLU networks of any width exceeding $\text{polylog}(n, 1/\epsilon, 1/\delta)$ suffice to achieve a test misclassification error of $\epsilon$. We also prove that stochastic gradient descent can achieve $\epsilon$ test error with polylogarithmic width and $\widetilde{\Theta}(1/\epsilon)$ samples. The analysis relies upon the separation margin of the limiting kernel, which is guaranteed positive, can distinguish between true labels and random labels, and can give a tight sample-complexity analysis in the infinite-width setting.

## 1 Introduction

Despite the extensive empirical success of deep networks, their optimization and generalization properties are still not fully understood. Recently, the neural tangent kernel (NTK) has provided the following insight into the problem. In the infinite-width limit, the NTK converges to a limiting kernel which stays constant during training; on the other hand, when the width is large enough, the function learned by gradient descent follows the NTK (Jacot et al., 2018). This motivates the study of overparameterized networks trained by gradient descent, using properties of the NTK. In fact, parameters related to the NTK, such as the minimum eigenvalue of the limiting kernel, appear to affect optimization and generalization (Arora et al., 2019).

However, in addition to such NTK-dependent parameters, prior work also requires the width to depend polynomially on $n$, $1/\delta$ or $1/\epsilon$, where $n$ denotes the size of the training set, $\delta$ denotes the failure probability, and $\epsilon$ denotes the target error. These large widths far exceed what is used empirically, constituting a significant gap between theory and practice.

**Our contributions.** In this paper, we narrow this gap by showing that a two-layer ReLU network with $\Omega(\ln(n/\delta) + \ln(1/\epsilon)^2)$ hidden units trained by gradient descent achieves classification error $\epsilon$ *on test data*, meaning both optimization and generalization occur. Unlike prior work, the width is fully polylogarithmic in $n$, $1/\delta$, and $1/\epsilon$; the width will additionally depend on the *separation margin* of the limiting kernel, a quantity which is guaranteed positive (assuming no inputs are parallel), can distinguish between true labels and random labels, and can give a tight sample-complexity analysis in the infinite-width setting. The paper organization together with some details are described below.

**Section 2** studies gradient descent on the training set. Using the $\ell_1$ geometry inherent in classification tasks, we prove that with any width at least polylogarithmic and any constant step size no larger than 1, gradient descent achieves training error $\epsilon$ in $\widetilde{\Theta}(1/\epsilon)$ iterations (cf. Theorem 2.2). As is common in the NTK literature (Chizat & Bach, 2019), we also show the parameters hardly change, which will be essential to our generalization analysis.

**Section 3** gives a test error bound. Concretely, using the preceding gradient descent analysis, and standard Rademacher tools and exploiting how little the weights moved, we show that with $\widetilde{\Omega}(1/\epsilon^2)$ samples and $\widetilde{\Theta}(1/\epsilon)$ iterations, gradient descent finds a solution with $\epsilon$ test error (cf. Theorem 3.2 and Corollary 3.3). (As discussed in Remark 3.4, $\widetilde{\Omega}(1/\epsilon)$ samples also suffice via a smoothness-based generalization bound, at the expense of large constant factors.)

**Section 4** considers stochastic gradient descent (SGD) with access to a standard stochastic online oracle. We prove that with width at least polylogarithmic and $\widetilde{\Theta}(1/\epsilon)$ samples, SGD achieves an arbitrarily small test error (cf. Theorem 4.1).

**Section 5** discusses the separation margin, which is in general a positive number, but reflects the difficulty of the classification problem in the infinite-width limit. While this margin can degrade all the way down to $O(1/\sqrt{n})$ for random labels, it can be much larger when there is a strong relationship between features and labels: for example, on the *noisy 2-XOR* data introduced in (Wei et al., 2018), we show that the margin is $\Omega(1/\ln(n))$, and our SGD sample complexity is tight in the infinite-width case.

**Section 6** concludes with some open problems.

## 1.1 RELATED WORK

There has been a large literature studying gradient descent on overparameterized networks via the NTK. The most closely related work is (Nitanda & Suzuki, 2019), which shows that a two-layer network trained by gradient descent with the logistic loss can achieve a small test error, under the same assumption that the NTK with respect to the first layer can separate the data distribution. However, they analyze smooth activations, while we handle the ReLU. They require $\Omega(1/\epsilon^2)$ hidden units, $\widetilde{\Omega}(1/\epsilon^4)$ data samples, and $O(1/\epsilon^2)$ steps, while our result only needs polylogarithmic hidden units, $\widetilde{\Omega}(1/\epsilon^2)$ data samples, and $\widetilde{O}(1/\epsilon)$ steps.

Additionally on shallow networks, Du et al. (2018b) prove that on an overparameterized two-layer network, gradient descent can globally minimize the empirical risk with the squared loss. Their result requires $\Omega(n^6/\delta^3)$ hidden units. Oymak & Soltanolkotabi (2019); Song & Yang (2019) further reduce the required overparameterization, but there is still a $\mathrm{poly}(n)$ dependency. Using the same amount of overparameterization as (Du et al., 2018b), Arora et al. (2019) further show that the two-layer network learned by gradient descent can achieve a small test error, assuming that on the data distribution the smallest eigenvalue of the limiting kernel is at least some positive constant. They also give a fine-grained characterization of the predictions made by gradient descent iterates; such a characterization makes use of a special property of the squared loss and cannot be applied to the logistic regression setting. Li & Liang (2018) show that stochastic gradient descent (SGD) with the cross entropy loss can learn a two-layer network with small test error, using $\mathrm{poly}(\ell, 1/\epsilon)$ hidden units, where $\ell$ is at least the covering number of the support of the feature distribution using balls whose radii are no larger than the smallest distance between two data points with different labels. Allen-Zhu et al. (2018a) consider SGD on a two-layer network, and a variant of SGD on a three-layer network. The three-layer analysis further exhibits some properties not captured by the NTK. They assume a ground truth network with infinite-order smooth activations, and they require the width to depend polynomially on $1/\epsilon$ and some constants related to the smoothness of the activations of the ground truth network.

On deep networks, a variety of works have established low training error (Allen-Zhu et al., 2018b; Du et al., 2018a; Zou et al., 2018; Zou & Gu, 2019). Allen-Zhu et al. (2018c) show that SGD can minimize the regression loss for recurrent neural networks, and Allen-Zhu & Li (2019b) further prove a low generalization error. Allen-Zhu & Li (2019a) show that using the same number of training examples, a three-layer ResNet can learn a function class with a much lower test error than any kernel method. Cao & Gu (2019a) assume that the NTK with respect to the second layer of a two-layer network can separate the data distribution, and prove that gradient descent on a deep network can achieve $\epsilon$ test error with $\Omega(1/\epsilon^4)$ samples and $\Omega(1/\epsilon^{14})$ hidden units. Cao & Gu (2019b) consider SGD with an online oracle and give a general result. Under the same assumption as in (Cao & Gu, 2019a), their result requires $\Omega(1/\epsilon^{14})$ hidden units and sample complexity $\widetilde{O}(1/\epsilon^2)$.

By contrast, with the same online oracle, our result only needs polylogarithmic hidden units and sample complexity $\widetilde{O}(1/\epsilon)$.

## 1.2 NOTATION

The dataset is denoted by $\{(x_i, y_i)\}_{i=1}^n$ where $x_i \in \mathbb{R}^d$ and $y_i \in \{-1, +1\}$. For simplicity, we assume that $\|x_i\|_2 = 1$ for any $1 \le i \le n$, which is standard in the NTK literature.

The two-layer network has weight matrices $W \in \mathbb{R}^{m \times d}$ and $a \in \mathbb{R}^m$. We use the following parameterization, which is also used in (Du et al., 2018b; Arora et al., 2019):

$$f(x; W, a) := \frac{1}{\sqrt{m}} \sum_{s=1}^m a_s \sigma\left(\langle w_s, x \rangle\right),$$

with initialization

$$w_{s,0} \sim \mathcal{N}(0, I_d), \quad \text{and} \quad a_s \sim \text{unif}\left(\{-1, +1\}\right).$$

Note that in this paper, $w_{s,t}$ denotes the $s$-th row of $W$ at step $t$. We fix $a$ and only train $W$, as in (Li & Liang, 2018; Du et al., 2018b; Arora et al., 2019; Nitanda & Suzuki, 2019). We consider the ReLU activation $\sigma(z) := \max\{0, z\}$, though our analysis can be extended easily to Lipschitz continuous, positively homogeneous activations such as leaky ReLU.

We use the logistic (binary cross entropy) loss $\ell(z) := \ln\left(1 + \exp(-z)\right)$ and gradient descent. For any $1 \le i \le n$ and any $W$, let $f_i(W) := f(x_i; W, a)$. The empirical risk and its gradient are given by

$$\widehat{\mathcal{R}}(W) := \frac{1}{n} \sum_{i=1}^n \ell\left(y_i f_i(W)\right), \quad \text{and} \quad \nabla \widehat{\mathcal{R}}(W) = \frac{1}{n} \sum_{i=1}^n \ell'\left(y_i f_i(W)\right) y_i \nabla f_i(W).$$

For any $t \ge 0$, the gradient descent step is given by $W_{t+1} := W_t - \eta_t \nabla \widehat{\mathcal{R}}(W_t)$. Also define

$$f_i^{(t)}(W) := \left\langle \nabla f_i(W_t), W \right\rangle, \quad \text{and} \quad \widehat{\mathcal{R}}^{(t)}(W) := \frac{1}{n} \sum_{i=1}^n \ell\left(y_i f_i^{(t)}(W)\right).$$

Note that $f_i^{(t)}(W_t) = f_i(W_t)$. This property generally holds due to homogeneity: for any $W$ and any $1 \le s \le m$,

$$\frac{\partial f_i}{\partial w_s} = \frac{1}{\sqrt{m}} a_s \mathbb{1}\left[\langle w_s, x_i \rangle > 0\right] x_i, \quad \text{and} \quad \left\langle \frac{\partial f_i}{\partial w_s}, w_s \right\rangle = \frac{1}{\sqrt{m}} a_s \sigma\left(\langle w_s, x_i \rangle\right),$$

and thus $\left\langle \nabla f_i(W), W \right\rangle = f_i(W)$.

## 2 EMPIRICAL RISK MINIMIZATION

In this section, we consider a fixed training set and empirical risk minimization. We first state our assumption on the separability of the NTK, and then give our main result and a proof sketch.

The key idea of the NTK is to do the first-order Taylor approximation:

$$f(x; W, a) \approx f(x; W_0, a) + \left\langle \nabla_W f(x; W_0, a), W - W_0 \right\rangle.$$

In other words, we want to do learning using the features given by $\nabla f_i(W_0) \in \mathbb{R}^{m \times d}$. A natural assumption is that there exists $\overline{U} \in \mathbb{R}^{m \times d}$ which can separate $\left\{\left(\nabla f_i(W_0), y_i\right)\right\}_{i=1}^n$ with a positive margin:

$$\min_{1 \le i \le n} \left( y_i \left\langle \overline{U}, \nabla f_i(W_0) \right\rangle \right) = \min_{1 \le i \le n} \left( y_i \frac{1}{\sqrt{m}} \sum_{s=1}^m a_s \langle \bar{u}_s, x_i \rangle \mathbb{1}\left[\langle w_{s,0}, x_i \rangle > 0\right] \right) > 0. \quad (2.1)$$

The infinite-width limit of eq. (2.1) is formalized as Assumption 2.1, with an additional bound on the $(2, \infty)$ norm of the separator. A concrete construction of $\overline{U}$ using Assumption 2.1 is given in eq. (2.2).

Let $\mu_{\mathcal{N}}$ denote the Gaussian measure on $\mathbb{R}^d$, given by the Gaussian density with respect to the Lebesgue measure on $\mathbb{R}^d$. We consider the following Hilbert space

$$\mathcal{H} := \left\{ w : \mathbb{R}^d \to \mathbb{R}^d \; \middle| \; \int \|w(z)\|_2^2 \, \mathrm{d}\mu_{\mathcal{N}}(z) < \infty \right\}.$$

For any $x \in \mathbb{R}^d$, define $\phi_x \in \mathcal{H}$ by

$$\phi_x(z) := x \mathbb{1}\left[ \langle z, x \rangle > 0 \right],$$

and particularly define $\phi_i := \phi_{x_i}$ for the training input $x_i$.

**Assumption 2.1.** There exists $\bar{v} \in \mathcal{H}$ and $\gamma > 0$, such that $\|\bar{v}(z)\|_2 \leq 1$ for any $z \in \mathbb{R}^d$, and for any $1 \leq i \leq n$,

$$y_i \langle \bar{v}, \phi_i \rangle_{\mathcal{H}} := y_i \int \langle \bar{v}(z), \phi_i(z) \rangle \, \mathrm{d}\mu_{\mathcal{N}}(z) \geq \gamma.$$

$$\Diamond$$

As discussed in Section 5, the space $\mathcal{H}$ is the reproducing kernel Hilbert space (RKHS) induced by the infinite-width NTK with respect to $W$, and $\phi_x$ maps $x$ into $\mathcal{H}$. Assumption 2.1 supposes that the induced training set $\{(\phi_i, y_i)\}_{i=1}^n$ can be separated by some $\bar{v} \in \mathcal{H}$, with an additional bound on $\|\bar{v}(z)\|_2$ which is crucial in our analysis. It is also possible to give a dual characterization of the separation margin (cf. eq. (5.2)), which also allows us to show that Assumption 2.1 always holds when there are no parallel inputs (cf. Proposition 5.1). However, it is often more convenient to construct $\bar{v}$ directly; see Section 5 for some examples.

With Assumption 2.1, we state our main empirical risk result.

**Theorem 2.2.** *Under Assumption 2.1, given any risk target $\epsilon \in (0, 1)$ and any $\delta \in (0, 1/3)$, let*

$$\lambda := \frac{\sqrt{2 \ln(4n/\delta)} + \ln(4/\epsilon)}{\gamma/4}, \quad \text{and} \quad M := \frac{4096\lambda^2}{\gamma^6}.$$

*Then for any $m \geq M$ and any constant step size $\eta \leq 1$, with probability $1 - 3\delta$ over the random initialization,*

$$\frac{1}{T} \sum_{t < T} \widehat{\mathcal{R}}(W_t) \leq \epsilon, \quad \text{where} \quad T := \lceil 2\lambda^2/\eta\epsilon \rceil.$$

*Moreover for any $0 \leq t < T$ and any $1 \leq s \leq m$,*

$$\left\| w_{s,t} - w_{s,0} \right\|_2 \leq \frac{4\lambda}{\gamma\sqrt{m}}.$$

While the number of hidden units required by prior work all have a polynomial dependency on $n$, $1/\delta$ or $1/\epsilon$, Theorem 2.2 only requires $m = \Omega\left(\ln(n/\delta) + \ln(1/\epsilon)^2\right)$. The required width has a polynomial dependency on $1/\gamma$, which is an adaptive quantity: while $1/\gamma$ can be $\mathrm{poly}(n)$ for random labels (cf. Proposition 5.2), it can be $\mathrm{polylog}(n)$ when there is a strong feature-label relationship, for example on the noisy 2-XOR data introduced in (Wei et al., 2018) (cf. Proposition 5.3). Moreover, we show in Proposition 5.4 that if we want $\left\{ \left( \nabla f_i(W_0), y_i \right) \right\}_{i=1}^n$ to be separable, which is the starting point of an NTK-style analysis, the width has to depend polynomially on $1/\gamma$.

In the rest of Section 2, we give a proof sketch of Theorem 2.2. The full proof is given in Appendix A.

## 2.1 PROPERTIES AT INITIALIZATION

In this subsection, we give some nice properties of random initialization.

Given an initialization $(W_0, a)$, for any $1 \leq s \leq m$, define

$$\bar{u}_s := \frac{1}{\sqrt{m}} a_s \bar{v}(w_{s,0}), \tag{2.2}$$

where $\bar{v}$ is given by Assumption 2.1. Collect $\bar{u}_s$ into a matrix $\overline{U} \in \mathbb{R}^{m \times d}$. It holds that $\|\bar{u}_s\|_2 \leq 1/\sqrt{m}$, and $\|\overline{U}\|_F \leq 1$.

Lemma 2.3 ensures that with high probability $\overline{U}$ has a positive margin at initialization.

**Lemma 2.3.** *Under Assumption 2.1, given any* $\delta \in (0,1)$ *and any* $\epsilon_1 \in (0,\gamma)$, *if* $m \geq \left(2\ln(n/\delta)\right)/\epsilon_1^2$, *then with probability* $1 - \delta$, *it holds simultaneously for all* $1 \leq i \leq n$ *that*

$$y_i f_i^{(0)}\left(\overline{U}\right) = y_i \left\langle \nabla f_i(W_0), \overline{U} \right\rangle \geq \gamma - \sqrt{\frac{2\ln(n/\delta)}{m}} \geq \gamma - \epsilon_1.$$

For any $W$, any $\epsilon_2 > 0$, and any $1 \leq i \leq n$, define

$$\alpha_i(W, \epsilon_2) = \frac{1}{m} \sum_{s=1}^{m} \mathbb{1}\left[\left|\langle w_s, x_i \rangle\right| \leq \epsilon_2\right].$$

Lemma 2.4 controls $\alpha_i(W_0, \epsilon_2)$. It will help us show that $\overline{U}$ has a good margin during the training process.

**Lemma 2.4.** *Under the condition of Lemma 2.3, for any* $\epsilon_2 > 0$, *with probability* $1 - \delta$, *it holds simultaneously for all* $1 \leq i \leq n$ *that*

$$\alpha_i\left(W_0, \epsilon_2\right) \leq \sqrt{\frac{2}{\pi}}\epsilon_2 + \sqrt{\frac{\ln(n/\delta)}{2m}} \leq \epsilon_2 + \frac{\epsilon_1}{2}.$$

Finally, Lemma 2.5 controls the output of the network at initialization.

**Lemma 2.5.** *Given any* $\delta \in (0,1)$, *if* $m \geq 25\ln(2n/\delta)$, *then with probability* $1 - \delta$, *it holds simultaneously for all* $1 \leq i \leq n$ *that*

$$\left|f(x_i; W_0, a)\right| \leq \sqrt{2\ln\left(4n/\delta\right)}.$$

## 2.2 Convergence analysis of gradient descent

We analyze gradient descent in this subsection. First, define

$$\widehat{\mathcal{Q}}(W) := \frac{1}{n} \sum_{i=1}^{n} -\ell'\left(y_i f_i(W)\right).$$

We have the following observations.

- For any $W$ and any $1 \leq s \leq m$, $\left\|\partial f_i/\partial w_s\right\|_2 \leq 1/\sqrt{m}$, and thus $\left\|\nabla f_i(W)\right\|_F \leq 1$. Therefore by the triangle inequality, $\left\|\nabla\widehat{\mathcal{R}}(W)\right\|_F \leq \widehat{\mathcal{Q}}(W)$.

- The logistic loss satisfies $0 \leq -\ell' \leq 1$, and thus $0 \leq \widehat{\mathcal{Q}}(W) \leq 1$.

- The logistic loss satisfies $-\ell' \leq \ell$, and thus $\widehat{\mathcal{Q}}(W) \leq \widehat{\mathcal{R}}(W)$.

The quantity $\widehat{\mathcal{Q}}$ first appeared in the perceptron analysis (Novikoff, 1962) for the ReLU loss, and has also been analyzed in prior work (Ji & Telgarsky, 2018; Cao & Gu, 2019a; Nitanda & Suzuki, 2019). In this work, $\widehat{\mathcal{Q}}$ specifically helps us prove the following result, which plays an important role in obtaining a width which only depends on $\mathrm{polylog}(1/\epsilon)$.

**Lemma 2.6.** *For any* $t \geq 0$ *and any* $\overline{W}$, *if* $\eta_t \leq 1$, *then*

$$\eta_t\widehat{\mathcal{R}}(W_t) \leq \left\|W_t - \overline{W}\right\|_F^2 - \left\|W_{t+1} - \overline{W}\right\|_F^2 + 2\eta_t\widehat{\mathcal{R}}^{(t)}\left(\overline{W}\right).$$

*Consequently, if we use a constant step size* $\eta \leq 1$ *for* $0 \leq \tau < t$, *then*

$$\eta\left(\sum_{\tau<t}\widehat{\mathcal{R}}(W_\tau)\right) + \left\|W_t - \overline{W}\right\|_F^2 \leq \left\|W_0 - \overline{W}\right\|_F^2 + 2\eta\left(\sum_{\tau<t}\widehat{\mathcal{R}}^{(\tau)}\left(\overline{W}\right)\right).$$

The proof of Lemma 2.6 starts from the standard iteration guarantee:

$$\left\|W_{t+1} - \overline{W}\right\|_F^2 = \left\|W_t - \overline{W}\right\|_F^2 - 2\eta_t \left\langle \nabla\widehat{\mathcal{R}}(W_t), W_t - \overline{W}\right\rangle + \eta_t^2 \left\|\nabla\widehat{\mathcal{R}}(W_t)\right\|_F^2.$$

We can then handle the inner product term using the convexity of $\ell$ and homogeneity of ReLU, and control $\|\nabla\widehat{\mathcal{R}}(W_t)\|_F^2$ by $\widehat{\mathcal{R}}(W_t)$ using the above properties of $\widehat{\mathcal{Q}}(W_t)$. Lemma 2.6 is similar to (Allen-Zhu & Li, 2019a, Fact D.4 and Claim D.5), where the squared loss is considered.

Using Lemmas 2.3 to 2.6, we can prove Theorem 2.2. Below is a proof sketch; the full proof is given in Appendix A.

1. We first show that as long as $\|w_{s,t} - w_{s,0}\|_2 \le 4\lambda/(\gamma\sqrt{m})$ for all $1 \le s \le m$, it holds that $\widehat{\mathcal{R}}^{(t)}\left(W_0 + \lambda\overline{U}\right) \le \epsilon/4$. To see this, let us consider $\widehat{\mathcal{R}}^{(0)}$ first. For any $1 \le i \le n$, Lemma 2.5 ensures that $|\langle\nabla f_i(W_0), W_0\rangle|$ is bounded, while Lemma 2.3 ensures that $\langle\nabla f_i(W_0), \overline{U}\rangle$ is concentrated around $\gamma$ with a large width. As a result, with the chosen $\lambda$ in Theorem 2.2, we can show that $\langle\nabla f_i(W_0), W_0 + \lambda\overline{U}\rangle$ is large, and $\widehat{\mathcal{R}}^{(0)}(W_0 + \lambda\overline{U})$ is small due to the exponential tail of the logistic loss. To further handle $\widehat{\mathcal{R}}^{(t)}$, we use a standard NTK argument to control $\langle\nabla f_i(W_t) - \nabla f_i(W_0), W_0 + \lambda\overline{U}\rangle$ under the condition that $\|w_{s,t} - w_{s,0}\|_2 \le 4\lambda/(\gamma\sqrt{m})$.

2. We then prove by contradiction that the above bound on $\|w_{s,t} - w_{s,0}\|_2$ holds for at least the first $T$ iterations. The key observation is that as long as $\widehat{\mathcal{R}}^{(t)}(W_0 + \lambda\overline{U}) \le \epsilon/4$, we can use it and Lemma 2.6 to control $\sum_{\tau<t} \widehat{\mathcal{Q}}(W_\tau)$, and then just invoke $\|w_{s,t} - w_{s,0}\|_2 \le \eta \sum_{\tau<t} \widehat{\mathcal{Q}}(W_\tau)/\sqrt{m}$.

   The quantity $\sum_{\tau<t} \widehat{\mathcal{Q}}(W_\tau)$ has also been considered in prior work (Cao & Gu, 2019a; Nitanda & Suzuki, 2019), where it is bounded by $\sqrt{t}\sqrt{\sum_{\tau<t}\widehat{\mathcal{Q}}(W_\tau)^2}$ using the Cauchy-Schwarz inequality, which introduces a $\sqrt{t}$ factor. To make the required width depend only on $\mathrm{polylog}(1/\epsilon)$, we also need an upper bound on $\sum_{\tau<t}\widehat{\mathcal{Q}}(W_\tau)$ which depends only on $\mathrm{polylog}(1/\epsilon)$. Since the above analysis results in a $\sqrt{t}$ factor, and in our case $\Omega(1/\epsilon)$ steps are needed, it is unclear how to get a $\mathrm{polylog}(1/\epsilon)$ width using the analysis in (Cao & Gu, 2019a; Nitanda & Suzuki, 2019). By contrast, using Lemma 2.6, we can show that $\sum_{\tau<t}\widehat{\mathcal{Q}}(W_\tau) \le 4\lambda/\gamma$, which only depends on $\ln(1/\epsilon)$.

3. The claims of Theorem 2.2 then follow directly from the above two steps and Lemma 2.6.

## 3 GENERALIZATION

To get a generalization bound, we naturally extend Assumption 2.1 to the following assumption.

**Assumption 3.1.** There exists $\bar{v} \in \mathcal{H}$ and $\gamma > 0$, such that $\left\|\bar{v}(z)\right\|_2 \le 1$ for any $z \in \mathbb{R}^d$, and

$$y \int \langle\bar{v}(z), x\rangle \mathbb{1}\left[\langle z, x\rangle > 0\right] \mathrm{d}\mu_{\mathcal{N}}(z) \ge \gamma$$

for almost all $(x, y)$ sampled from the data distribution $\mathcal{D}$. $\diamondsuit$

The above assumption is also made in (Nitanda & Suzuki, 2019) for smooth activations. (Cao & Gu, 2019a) make a similar separability assumption, but in the RKHS induced by the second layer $a$; by contrast, Assumption 3.1 is on separability in the RKHS induced by the first layer $W$.

Here is our test error bound with Assumption 3.1.

**Theorem 3.2.** *Under Assumption 3.1, given any $\epsilon \in (0,1)$ and any $\delta \in (0,1/4)$, let $\lambda$ and $M$ be given as in Theorem 2.2:*

$$\lambda := \frac{\sqrt{2\ln(4n/\delta)} + \ln(4/\epsilon)}{\gamma/4}, \quad and \quad M := \frac{4096\lambda^2}{\gamma^6}.$$

*Then for any $m \geq M$ and any constant step size $\eta \leq 1$, with probability $1 - 4\delta$ over the random initialization and data sampling,*

$$P_{(x,y)\sim\mathcal{D}} \left( yf(x; W_k, a) \leq 0 \right) \leq 2\epsilon + \frac{16 \left( \sqrt{2\ln(4n/\delta)} + \ln(4/\epsilon) \right)}{\gamma^2 \sqrt{n}} + 6\sqrt{\frac{\ln(2/\delta)}{2n}},$$

*where $k$ denotes the step with the minimum empirical risk before $\lceil 2\lambda^2/\eta\epsilon \rceil$.*

Below is a direct corollary of Theorem 3.2.

**Corollary 3.3.** *Under Assumption 3.1, given any $\epsilon, \delta \in (0,1)$, using a constant step size no larger than 1 and let*

$$n = \widetilde{\Omega} \left( \frac{1}{\gamma^4 \epsilon^2} \right), \quad and \quad m = \Omega \left( \frac{\ln(n/\delta) + \ln(1/\epsilon)^2}{\gamma^8} \right),$$

*it holds with probability $1 - \delta$ that $P_{(x,y)\sim\mathcal{D}} \left( yf(x; W_k, a) \leq 0 \right) \leq \epsilon$, where $k$ denotes the step with the minimum empirical risk in the first $\widetilde{\Theta}(1/\gamma^2\epsilon)$ steps.*

The proof of Theorem 3.2 uses the sigmoid mapping $-\ell'(z) = e^{-z}/(1+e^{-z})$, the empirical average $\widehat{\mathcal{Q}}(W_k)$, and the corresponding population average $\mathcal{Q}(W_k) := \mathbb{E}_{(x,y)\sim\mathcal{D}} \left[ -\ell' \left( yf(x; W_k, a) \right) \right]$. As noted in (Cao & Gu, 2019a), because $P_{(x,y)\sim\mathcal{D}} \left( yf(x; W_k, a) \leq 0 \right) \leq 2\mathcal{Q}(W_k)$, it is enough to control $\mathcal{Q}(W_k)$. As $\widehat{\mathcal{Q}}(W_k)$ is controlled by Theorem 2.2, it is enough to control the generalization error $\mathcal{Q}(W_k) - \widehat{\mathcal{Q}}(W_k)$. Moreover, since $-\ell'$ is supported on $[0,1]$ and 1-Lipschitz, it is enough to bound the Rademacher complexity of the function space explored by gradient descent. Invoking the bound on $\left\| W_k^\top - W_0^\top \right\|_{2,\infty}$ finishes the proof. The proof details are given in Appendix B.

**Remark 3.4.** To get Theorem 3.2, we use a Lipschitz-based Rademacher complexity bound. One can also use a smoothness-based Rademacher complexity bound (Srebro et al., 2010, Theorem 1) and get a sample complexity $\widetilde{O}(1/\gamma^4\epsilon)$. However, the bound will become complicated and some large constant will be introduced. It is an interesting open question to give a clean analysis based on smoothness. $\diamondsuit$

## 4 STOCHASTIC GRADIENT DESCENT

There are some different formulations of SGD. In this section, we consider SGD with an online oracle. We randomly sample $W_0$ and $a$, and fix $a$ during training. At step $i$, a data example $(x_i, y_i)$ is sampled from the data distribution. We still let $f_i(W) := f(x_i; W, a)$, and perform the following update

$$W_{i+1} := W_i - \eta_i \ell' \left( y_i f_i(W_i) \right) y_i \nabla f_i(W_i).$$

Note that here $i$ starts from 0.

Still with Assumption 3.1, we show the following result.

**Theorem 4.1.** *Under Assumption 3.1, given any $\epsilon, \delta \in (0,1)$, using a constant step size and $m = \Omega \left( (\ln(1/\delta) + \ln(1/\epsilon)^2)/\gamma^8 \right)$, it holds with probability $1 - \delta$ that*

$$\frac{1}{n} \sum_{i=1}^{n} P_{(x,y)\sim\mathcal{D}} \left( yf(x; W_i, a) \leq 0 \right) \leq \epsilon, \quad for \quad n = \widetilde{\Theta}(1/\gamma^2\epsilon).$$

Below is a proof sketch of Theorem 4.1; the complete proof is given in Appendix C. For any $i$ and $W$, define

$$\mathcal{R}_i(W) := \ell \left( y_i \left\langle \nabla f_i(W_i), W \right\rangle \right), \quad and \quad \mathcal{Q}_i(W) := -\ell' \left( y_i \left\langle \nabla f_i(W_i), W \right\rangle \right).$$

Due to homogeneity, it holds that $\mathcal{R}_i(W_i) = \ell \left( y_i f_i(W_i) \right)$ and $\mathcal{Q}_i(W_i) = -\ell' \left( y_i f_i(W_i) \right)$.

The first step is an extension of Lemma 2.6 to the SGD setting, with a similar proof.

**Lemma 4.2.** *With a constant step size $\eta \leq 1$, for any $\overline{W}$ and any $i \geq 0$,*

$$\eta \left( \sum_{t < i} \mathcal{R}_t(W_t) \right) + \left\| W_i - \overline{W} \right\|_F^2 \leq \left\| W_0 - \overline{W} \right\|_F^2 + 2\eta \left( \sum_{t < i} \mathcal{R}_t \left( \overline{W} \right) \right).$$

With Lemma 4.2, we can also extend Theorem 2.2 to the SGD setting and get a bound on $\sum_{i < n} \mathcal{Q}_i(W_i)$, using a similar proof. To further get a bound on the cumulative population risk $\sum_{i < n} \mathcal{Q}(W_i)$, the key observation is that $\sum_{i < n} \left( \mathcal{Q}(W_i) - \mathcal{Q}_i(W_i) \right)$ is a martingale. Using a martingale Bernstein bound, we prove the following lemma; applying it finishes the proof of Theorem 4.1.

**Lemma 4.3.** *Given any $\delta \in (0, 1)$, with probability $1 - \delta$,*

$$\sum_{t < i} \mathcal{Q}(W_t) \leq 4 \sum_{t < i} \mathcal{Q}_t(W_t) + 4 \ln \left( \frac{1}{\delta} \right).$$

## 5 ON SEPARABILITY

In this section we give some discussion on Assumption 2.1, the separability of the NTK. The proofs are all given in Appendix D.

Given a training set $\left\{ (x_i, y_i) \right\}_{i=1}^n$, the linear kernel is defined as $K_0(x_i, x_j) := \langle x_i, x_j \rangle$. The maximum margin achievable by a linear classifier is given by

$$\gamma_0 := \min_{q \in \Delta_n} \sqrt{(q \odot y)^\top K_0 (q \odot y)}. \tag{5.1}$$

where $\Delta_n$ denotes the probability simplex and $\odot$ denotes the Hadamard product. In addition to the dual definition eq. (5.1), when $\gamma_0 > 0$ there also exists a maximum margin classifier $\bar{u}$ which gives a primal characterization of $\gamma_0$: it holds that $\|\bar{u}\|_2 = 1$ and $y_i \langle \bar{u}, x_i \rangle \geq \gamma_0$ for all $i$.

In this paper we consider another kernel, the infinite-width NTK with respect to the first layer:

$$K_1(x_i, x_j) := \mathbb{E} \left[ \frac{\partial f(x_i; W_0, a)}{\partial W_0}, \frac{\partial f(x_j; W_0, a)}{\partial W_0} \right]$$

$$= \mathbb{E}_{w \sim \mathcal{N}(0, I_d)} \left[ \left\langle x_i \mathbb{1} \left[ \langle x_i, w \rangle > 0 \right], x_j \mathbb{1} \left[ \langle x_j, w \rangle > 0 \right] \right\rangle \right] = \langle \phi_i, \phi_j \rangle_{\mathcal{H}}.$$

Here $\phi$ and $\mathcal{H}$ are defined at the beginning of Section 2. Similar to the dual definition of $\gamma_0$, the margin given by $K_1$ is defined as

$$\gamma_1 := \min_{q \in \Delta_n} \sqrt{(q \odot y)^\top K_1 (q \odot y)}. \tag{5.2}$$

We can also give a primal characterization of $\gamma_1$ when it is positive.

**Proposition 5.1.** *If $\gamma_1 > 0$, then there exists $\hat{v} \in \mathcal{H}$ such that $\|\hat{v}\|_{\mathcal{H}} = 1$, and $y_i \langle \hat{v}, \phi_i \rangle_{\mathcal{H}} \geq \gamma_1$ for any $1 \leq i \leq n$. Additionally $\left\| \hat{v}(z) \right\|_2 \leq 1/\gamma_1$ for any $z \in \mathbb{R}^d$.*

The proof is given in Appendix D, and uses the Fenchel duality theory. Using the upper bound $\left\| \hat{v}(z) \right\|_2 \leq 1/\gamma_1$, we can see that $\gamma_1 \hat{v}$ satisfies Assumption 2.1 with $\gamma \geq \gamma_1^2$. However, such an upper bound $\left\| \hat{v}(z) \right\|_2 \leq 1/\gamma_1$ might be too loose, which leads to a bad rate. In fact, as shown later, in some cases we can construct $\bar{v}$ directly which satisfies Assumption 2.1 with a large $\gamma$. For this reason, we choose to make Assumption 2.1 instead of assuming a positive $\gamma_1$.

However, we can use $\gamma_1$ to show that Assumption 2.1 always holds when there are no parallel inputs. Oymak & Soltanolkotabi (2019, Corollary I.2) prove that if for any two feature vectors $x_i$ and $x_j$, we have $\|x_i - x_j\|_2 \geq \theta$ and $\|x_i + x_j\|_2 \geq \theta$ for some $\theta > 0$, then the minimum eigenvalue of $K_1$ is at least $\theta/(100n^2)$. For arbitrary labels $y \in \{-1, +1\}^n$, since $\|q \odot y\|_2 \geq 1/\sqrt{n}$, we have the worst case bound $\gamma_1^2 \geq \theta/100n^3$. A direct improvement of this bound is $\theta/100n_S^3$, where $n_S$ denotes the number of support vectors, which could be much smaller than $n$ with real world data.

On the other hand, given any training set $\left\{ (x_i, y_i) \right\}_{i=1}^n$ which may have a large margin, replacing $y$ with random labels would destroy the margin, which is what should be expected.

**Proposition 5.2.** *Given any training set $\{(x_i, y_i)\}_{i=1}^{n}$, if the true labels $y$ are replaced with random labels $\epsilon \sim \mathrm{unif}\left(\{-1, +1\}^n\right)$, then with probability $0.9$ over the random labels, it holds that $\gamma_1 \leq 1/\sqrt{20n}$.*

Although the above bounds all have a polynomial dependency on $n$, they hold for arbitrary or random labels, and thus do not assume any relationship between the features and labels. Next we give some examples where there is a strong feature-label relationship, and thus a much larger margin can be proved.

## 5.1 THE LINEARLY SEPARABLE CASE

Suppose the data distribution is linearly separable with margin $\gamma_0$: there exists a unit vector $\bar{u}$ such that $y \langle \bar{u}, x \rangle \geq \gamma_0$ almost surely. Then we can define $\bar{v}(z) := \bar{u}$ for any $z \in \mathbb{R}^d$. For almost all $(x, y)$, we have

$$y \int \langle \bar{v}(z), x \rangle \mathbb{1}\left[\langle z, x \rangle > 0\right] \mathrm{d}\mu_{\mathcal{N}}(z) = \int y \langle \bar{u}, x \rangle \mathbb{1}\left[\langle z, x \rangle > 0\right] \mathrm{d}\mu_{\mathcal{N}}(z)$$

$$\geq \gamma \int \mathbb{1}\left[\langle z, x \rangle > 0\right] \mathrm{d}\mu_{\mathcal{N}}(z)$$

$$= \frac{\gamma_0}{2},$$

and thus Assumption 2.1 holds with $\gamma = \gamma_0/2$.

## 5.2 THE NOISY 2-XOR DISTRIBUTION

We consider the noisy 2-XOR distribution introduced in (Wei et al., 2018). It is the uniform distribution over the following $2^d$ points:

$$(x_1, x_2, y, x_3, \ldots, x_d) \in \left\{\left(\frac{1}{\sqrt{d-1}}, 0, 1\right), \left(0, \frac{1}{\sqrt{d-1}}, -1\right), \left(\frac{-1}{\sqrt{d-1}}, 0, 1\right), \left(0, \frac{-1}{\sqrt{d-1}}, -1\right)\right\}$$

$$\times \left\{\frac{-1}{\sqrt{d-1}}, \frac{1}{\sqrt{d-1}}\right\}^{d-2}.$$

The factor $1/\sqrt{d-1}$ ensures that $\|x\|_2 = 1$, and $\times$ above denotes the Cartesian product. Here the label $y$ only depends on the first two coordinates of the input $x$.

To construct $\bar{v}$, we first decompose $\mathbb{R}^2$ into four regions:

$$A_1 := \left\{(z_1, z_2) \mid z_1 \geq 0, |z_1| \geq |z_2|\right\},$$
$$A_2 := \left\{(z_1, z_2) \mid z_2 > 0, |z_1| < |z_2|\right\},$$
$$A_3 := \left\{(z_1, z_2) \mid z_1 \leq 0, |z_1| \geq |z_2|\right\} \setminus \{(0, 0)\},$$
$$A_4 := \left\{(z_1, z_2) \mid z_2 < 0, |z_1| < |z_2|\right\}.$$

Then $\bar{v}$ can de defined as follows. It only depends on the first two coordinates of $z$.

$$\bar{v}(z) := \begin{cases} (1, 0, 0, \ldots, 0) & \text{if } (z_1, z_2) \in A_1, \\ (0, -1, 0, \ldots, 0) & \text{if } (z_1, z_2) \in A_2, \\ (-1, 0, 0, \ldots, 0) & \text{if } (z_1, z_2) \in A_3, \\ (0, 1, 0, \ldots, 0) & \text{if } (z_1, z_2) \in A_4. \end{cases} \tag{5.3}$$

The following result shows that $\gamma = \Omega(1/d)$. Note that $n$ could be as large as $2^d$, in which case $\gamma$ is basically $O\left(1/\ln(n)\right)$.

**Proposition 5.3.** *For any $(x, y)$ sampled from the noisy 2-XOR distribution and any $d \geq 3$, it holds that*

$$y \int \langle \bar{v}(z), x \rangle \mathbb{1}\left[\langle z, x \rangle > 0\right] \mathrm{d}\mu_{\mathcal{N}}(z) \geq \frac{1}{60d}.$$

We can prove two other interesting results for the noisy 2-XOR data.

**The width needs a** $\text{poly}(1/\gamma)$ **dependency for initial separability.** The first step of an NTK analysis is to show that $\left\{\left(\nabla f_i(W_0), y_i\right)\right\}_{i=1}^{n}$ is separable. Proposition 5.4 gives an example where $\left\{\left(\nabla f_i(W_0), y_i\right)\right\}_{i=1}^{n}$ is nonseparable when the network is narrow.

**Proposition 5.4.** *Let* $D = \{(x_i, y_i)\}_{i=1}^{4}$ *denote an arbitrary subset of the noisy 2-XOR dataset such that* $x_i$*'s have the same last* $(d-2)$ *coordinates. For any* $d \geq 20$*, if* $m \leq \sqrt{d-2}/4$*, then with probability* $1/2$ *over the random initialization of* $W_0$*, for any weights* $V \in \mathbb{R}^{m \times d}$*, it holds that* $y_i \langle V, \nabla f_i(W_0) \rangle \leq 0$ *for at least one* $i \in \{1, 2, 3, 4\}$*.*

For the noisy 2-XOR data, the separator $\bar{v}$ given by eq. (5.3) has margin $\gamma = \Omega(1/d)$, and $1/\gamma = O(d)$. As a result, if we want $\left\{\left(\nabla f_i(W_0), y_i\right)\right\}_{i=1}^{n}$ to be separable, the width has to be $\Omega(1/\sqrt{\gamma})$. For a smaller width, gradient descent might still be able to solve the problem, but a beyond-NTK analysis would be needed.

**A tight sample complexity upper bound for the infinite-width NTK.** (Wei et al., 2018) give a $d^2$ sample complexity lower bound for any NTK classifier on the noisy 2-XOR data. It turns out that $\gamma$ could give a *matching* sample complexity upper bound for the NTK and SGD.

(Wei et al., 2018) consider the infinite-width NTK with respect to both layers. For the first layer, the infinite-width NTK $K_1$ is defined in Section 5, and the corresponding RKHS $\mathcal{H}$ and RKHS mapping $\phi$ is defined in Section 2. For the second layer, the infinite width NTK is defined by

$$K_2\left(x_i, x_j\right) := \mathbb{E}\left[\frac{\partial f(x_i; W_0, a)}{\partial a}, \frac{\partial f(x_j; W_0, a)}{\partial a}\right]$$

$$= \mathbb{E}_{w \sim \mathcal{N}(0, I_d)}\left[\sigma\left(\langle w, x_i \rangle\right) \sigma\left(\langle w, x_j \rangle\right)\right].$$

The corresponding RKHS $\mathcal{K}$ and inner product $\langle w_1, w_2 \rangle_{\mathcal{K}}$ are given by

$$\mathcal{K} := \left\{w : \mathbb{R}^d \to \mathbb{R} \;\middle|\; \int w(z)^2 \, \mathrm{d}\mu_{\mathcal{N}}(z) < \infty\right\}, \quad \text{and} \quad \langle w_1, w_2 \rangle_{\mathcal{K}} = \int w_1(z) w_2(z) \, \mathrm{d}\mu_{\mathcal{N}}(z).$$

Given any $x \in \mathbb{R}^d$, it is mapped into $\psi_x \in \mathcal{K}$, where $\psi_x(z) := \sigma\left(\langle z, x \rangle\right)$. It holds that $K_2(x_i, x_j) = \langle \psi_{x_i}, \psi_{x_j} \rangle_{\mathcal{K}}$. The infinite-width NTK with respect to both layers is just $K_1 + K_2$. The corresponding RHKS is just $\mathcal{H} \times \mathcal{K}$ with the inner product

$$\langle (v_1, w_1), (v_2, w_2) \rangle_{\mathcal{H} \times \mathcal{K}} = \langle v_1, v_2 \rangle_{\mathcal{H}} + \langle w_1, w_2 \rangle_{\mathcal{K}}.$$

The classifier $\bar{v}$ considered in eq. (5.3) has a unit norm (i.e., $\|\bar{v}\|_{\mathcal{H}} = 1$) and margin $\gamma$ on the space $\mathcal{H}$. On $\mathcal{H} \times \mathcal{K}$, it is enough to consider $(\bar{v}, 0)$, which also has a unit norm and margin $\gamma$. Since the infinite-width NTK model is a linear model in $\mathcal{H} \times \mathcal{K}$, (Ji & Telgarsky, 2018, Lemma 2.5) can be used to show that SGD on the RKHS $\mathcal{H} \times \mathcal{K}$ could obtain a test error of $\epsilon$ with a sample complexity of $\widetilde{O}(1/\gamma^2 \epsilon)$. (The analysis in (Ji & Telgarsky, 2018) is done in $\mathbb{R}^d$, but it still works with a well-defined inner product.) Since $\gamma = \Omega(1/d)$, to achieve a constant test accuracy we need $\widetilde{O}(d^2)$ samples. This mathces (up to logarithmic factors) the sample complexity lower bound of $d^2$ given by Wei et al. (2018).

# 6 OPEN PROBLEMS

In this paper, we analyze gradient descent on a two-layer network in the NTK regime, where the weights stay close to the initialization. It is an interesting open question if gradient descent learns something beyond the NTK, after the iterates move far enough from the initial weights. It is also interesting to extend our analysis to other architectures, such as multi-layer networks, convolutional networks, and residual networks. Finally, in this paper we only discuss binary classification; it is interesting to see if it is possible to get similar results for other tasks, such as regression.

ACKNOWLEDGEMENTS

The authors are grateful for support from the NSF under grant IIS-1750051, and from NVIDIA via a GPU grant.

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

## A  OMITTED PROOFS FROM SECTION 2

*Proof of Lemma 2.3.* By Assumption 2.1, given any $1 \leq i \leq n$,

$$\mu := \mathbb{E}_{w \sim \mathcal{N}(0, I_d)} \left[ y_i \left\langle \bar{v}(w), x_i \right\rangle \mathbb{1} \left[ \left\langle w, x_i \right\rangle > 0 \right] \right] \geq \gamma.$$

On the other hand,

$$y_i f_i^{(0)} \left( \overline{U} \right) = \frac{1}{m} \sum_{s=1}^{m} y_i \left\langle \bar{v}(w_{s,0}), x_i \right\rangle \mathbb{1} \left[ \left\langle w_{s,0}, x_i \right\rangle > 0 \right]$$

is the empirical mean of i.i.d. r.v.'s supported on $[-1, +1]$ with mean $\mu$. Therefore by Hoeffding's inequality, with probability $1 - \delta/n$,

$$y_i f_i^{(0)} \left( \overline{U} \right) - \gamma \geq y_i f_i^{(0)} \left( \overline{U} \right) - \mu \geq -\sqrt{\frac{2 \ln(n/\delta)}{m}}.$$

Applying a union bound finishes the proof. $\qquad\square$

*Proof of Lemma 2.4.* Given any fixed $\epsilon_2$ and $1 \leq i \leq n$,

$$\mathbb{E} \left[ \alpha_i(W_0, \epsilon_2) \right] = \mathbb{P} \left( \left| \left\langle w, x_i \right\rangle \right| \leq \epsilon_2 \right) \leq \frac{2\epsilon_2}{\sqrt{2\pi}} = \sqrt{\frac{2}{\pi}} \epsilon_2,$$

because $\left\langle w, x_i \right\rangle$ is a standard Gaussian r.v. and the density of standard Gaussian has maximum $1/\sqrt{2\pi}$. Since $\alpha_i(W_0, \epsilon_2)$ is the empirical mean of Bernoulli r.v.'s, by Hoeffding's inequality, with probability $1 - \delta/n$,

$$\alpha_i(W_0, \epsilon_2) \leq \mathbb{E} \left[ \alpha_i(W_0, \epsilon_2) \right] + \sqrt{\frac{\ln(n/\delta)}{2m}} \leq \sqrt{\frac{2}{\pi}} \epsilon_2 + \sqrt{\frac{\ln(n/\delta)}{2m}}.$$

Applying a union bound finishes the proof. $\qquad\square$

To prove Lemma 2.5, we need the following technical result.

**Lemma A.1.** *Consider the random vector $X = (X_1, \ldots, X_m)$, where $X_i = \sigma(Z_i)$ for some $\sigma : \mathbb{R} \to \mathbb{R}$ that is 1-Lipschitz, and $Z_i$ are i.i.d. standard Gaussian r.v.'s. Then the r.v. $\|X\|_2$ is 1-sub-Gaussian, and thus with probability $1 - \delta$,*

$$\|X\|_2 - \mathbb{E}\left[\|X\|_2\right] \leq \sqrt{2\ln(1/\delta)}.$$

*Proof.* Given $a \in \mathbb{R}^m$, define

$$f(a) = \sqrt{\sum_{i=1}^{m} \sigma(a_i)^2} = \left\|\sigma(a)\right\|_2,$$

where $\sigma(a)$ is obtained by applying $\sigma$ coordinate-wisely to $a$. For any $a, b \in \mathbb{R}^m$, by the triangle inequality, we have

$$\left|f(a) - f(b)\right| = \left|\left\|\sigma(a)\right\|_2 - \left\|\sigma(b)\right\|_2\right| \leq \left\|\sigma(a) - \sigma(b)\right\|_2 = \sqrt{\sum_{i=1}^{m} \left(\sigma(a_i) - \sigma(b_i)\right)^2},$$

and by further using the 1-Lipschitz continuity of $\sigma$, we have

$$\left|f(a) - f(b)\right| \leq \sqrt{\sum_{i=1}^{m} \left(\sigma(a_i) - \sigma(b_i)\right)^2} \leq \sqrt{\sum_{i=1}^{m} (a_i - b_i)^2} = \|a - b\|_2.$$

As a result, $f$ is a 1-Lipschitz continuous function w.r.t. the $\ell_2$ norm, indeed $f(X)$ is 1-sub-Gaussian and the bound follows by Gaussian concentration (Wainwright, 2015, Theorem 2.4). $\square$

*Proof of Lemma 2.5.* Given $1 \leq i \leq n$, let $h_i = \sigma(W_0 x_i)/\sqrt{m}$. By Lemma A.1, $\|h_i\|_2$ is sub-Gaussian with variance proxy $1/m$, and with probability at least $1 - \delta/2n$ over $W_0$,

$$\|h_i\|_2 - \mathbb{E}\left[\|h_i\|_2\right] \leq \sqrt{\frac{2\ln(2n/\delta)}{m}} \leq \sqrt{\frac{2\ln(2n/\delta)}{25\ln(2n/\delta)}} \leq 1 - \frac{\sqrt{2}}{2}.$$

On the other hand, by Jensen's inequality,

$$\mathbb{E}\left[\|h_i\|_2\right] \leq \sqrt{\mathbb{E}\left[\|h_i\|_2^2\right]} = \frac{\sqrt{2}}{2}.$$

As a result, with probability $1 - \delta/2n$, it holds that $\|h_i\|_2 \leq 1$. By a union bound, with probability $1 - \delta/2$ over $W_0$, for all $1 \leq i \leq n$, we have $\|h_i\|_2 \leq 1$.

For any $W_0$ such that the above event holds, and for any $1 \leq i \leq n$, the r.v. $\langle h_i, a \rangle$ is sub-Gaussian with variance proxy $\|h_i\|_2^2 \leq 1$. By Hoeffding's inequality, with probability $1 - \delta/2n$ over $a$,

$$\left|\langle h_i, a \rangle\right| = \left|f(x_i; W_0, a)\right| \leq \sqrt{2\ln\left(4n/\delta\right)}.$$

By a union bound, with probability $1 - \delta/2$ over $a$, for all $1 \leq i \leq n$, we have $\left|f(x_i; W_0, a)\right| \leq \sqrt{2\ln\left(4n/\delta\right)}$.

The probability that the above events all happen is at least $(1 - \delta/2)(1 - \delta/2) \geq 1 - \delta$, over $W_0$ and $a$. $\square$

*Proof of Lemma 2.6.* We have

$$\left\|W_{t+1} - \overline{W}\right\|_F^2 = \left\|W_t - \overline{W}\right\|_F^2 - 2\eta_t \left\langle \nabla\widehat{\mathcal{R}}(W_t), W_t - \overline{W} \right\rangle + \eta_t^2 \left\|\nabla\widehat{\mathcal{R}}(W_t)\right\|_F^2. \tag{A.1}$$

The first order term of eq. (A.1) can be handled using the convexity of $\ell$ and homogeneity of ReLU:

$$
\begin{aligned}
\left\langle \nabla \widehat{\mathcal{R}}(W_t), W_t - \overline{W} \right\rangle &= \frac{1}{n} \sum_{i=1}^{n} \ell' \left( y_i f_i(W_t) \right) y_i \left\langle \nabla f_i(W_t), W_t - \overline{W} \right\rangle \\
&= \frac{1}{n} \sum_{i=1}^{n} \ell' \left( y_i f_i(W_t) \right) \left( y_i f_i(W_t) - y_i f_i^{(t)} \left( \overline{W} \right) \right) \\
&\geq \frac{1}{n} \sum_{i=1}^{n} \left( \ell \left( y_i f_i(W_t) \right) - \ell \left( y_i f_i^{(t)} \left( \overline{W} \right) \right) \right) = \widehat{\mathcal{R}}(W_t) - \widehat{\mathcal{R}}^{(t)} \left( \overline{W} \right).
\end{aligned}
$$
(A.2)

The second-order term of eq. (A.1) can be bounded as follows

$$
\eta_t^2 \left\| \nabla \widehat{\mathcal{R}}(W_t) \right\|_F^2 \leq \eta_t^2 \widehat{\mathcal{Q}}(W_t)^2 \leq \eta_t \widehat{\mathcal{Q}}(W_t) \leq \eta_t \widehat{\mathcal{R}}(W_t),
$$
(A.3)

because $\left\| \nabla \widehat{\mathcal{R}}(W_t) \right\|_F \leq \widehat{\mathcal{Q}}(W_t)$, and $\eta_t, \widehat{\mathcal{Q}}(W_t) \leq 1$, and $\widehat{\mathcal{Q}}(W_t) \leq \widehat{\mathcal{R}}(W_t)$. Combining eqs. (A.1) to (A.3) gives

$$
\eta_t \widehat{\mathcal{R}}(W_t) \leq \left\| W_t - \overline{W} \right\|_F^2 - \left\| W_{t+1} - \overline{W} \right\|_F^2 + 2\eta_t \widehat{\mathcal{R}}^{(t)} \left( \overline{W} \right).
$$

Telescoping gives the other claim. $\qquad\square$

*Proof of Theorem 2.2.* The required width ensures that with probability $1 - 3\delta$, Lemmas 2.3 to 2.5 hold with $\epsilon_1 = \gamma^2/8$ and $\epsilon_2 = 4\lambda/(\gamma\sqrt{m})$.

Let $t_1$ denote the first step such that there exists $1 \leq s \leq m$ with $\left\| w_{s,t_1} - w_{s,0} \right\|_2 > 4\lambda/(\gamma\sqrt{m})$. Therefore for any $0 \leq t < t_1$ and any $1 \leq s \leq m$, it holds that $\left\| w_{s,t} - w_{s,0} \right\|_2 \leq 4\lambda/(\gamma\sqrt{m})$. In addition, we let $\overline{W} := W_0 + \lambda \overline{U}$.

We first prove that for any $0 \leq t < t_1$, it holds that $\widehat{\mathcal{R}}^{(t)} \left( \overline{W} \right) \leq \epsilon/4$. Since $\ln(1 + r) \leq r$ for any $r$, the logistic satisfies $\ell(z) = \ln(1 + \exp(-z)) \leq \exp(-z)$, and it is enough to prove that for any $1 \leq i \leq n$,

$$
y_i \left\langle \nabla f_i(W_t), \overline{W} \right\rangle \geq \ln \left( \frac{4}{\epsilon} \right).
$$

We will split the left hand side into three terms and control them individually:

$$
y_i \left\langle \nabla f_i(W_t), \overline{W} \right\rangle = y_i \left\langle \nabla f_i(W_0), W_0 \right\rangle + y_i \left\langle \nabla f_i(W_t) - \nabla f_i(W_0), W_0 \right\rangle + \lambda y_i \left\langle \nabla f_i(W_t), \overline{U} \right\rangle.
$$
(A.4)

- The first term of eq. (A.4) can be controlled using Lemma 2.5:

$$
\left| y_i \left\langle \nabla f_i(W_0), W_0 \right\rangle \right| \leq \sqrt{2 \ln(4n/\delta)}.
$$
(A.5)

- The second term of eq. (A.4) can be written as

$$
y_i \left\langle \nabla f_i(W_t) - \nabla f_i(W_0), W_0 \right\rangle = y_i \frac{1}{\sqrt{m}} \sum_{s=1}^{m} a_s \left( \mathbb{1} \left[ \langle w_{s,t}, x_i \rangle > 0 \right] - \mathbb{1} \left[ \langle w_{s,0}, x_i \rangle > 0 \right] \right) \langle w_{s,0}, x_i \rangle.
$$

Let $S_c := \left\{ s \mid \mathbb{1} \left[ \langle w_{s,t}, x_i \rangle > 0 \right] - \mathbb{1} \left[ \langle w_{s,0}, x_i \rangle > 0 \right] \neq 0, 1 \leq s \leq m \right\}$. Note that $s \in S_c$ implies

$$
\left| \langle w_{s,0}, x_i \rangle \right| \leq \left| \langle w_{s,t} - w_{s,0}, x_i \rangle \right| \leq \left\| w_{s,t} - w_{s,0} \right\|_2 \left\| x_i \right\|_2 = \left\| w_{s,t} - w_{s,0} \right\|_2 \leq 4\lambda/(\gamma\sqrt{m}) = \epsilon_2.
$$

Therefore Lemma 2.4 ensures that

$$|S_c| \leq \left| \left\{ s \ \middle| \ |\langle w_{s,0}, x_i \rangle| \leq \epsilon_2 \right\} \right| \leq m \left( \frac{4\lambda}{\gamma\sqrt{m}} + \frac{\epsilon_1}{2} \right) = m \left( \frac{4\lambda}{\gamma\sqrt{m}} + \frac{\gamma^2}{16} \right).$$

and thus

$$\left| y_i \left\langle \nabla f_i(W_t) - \nabla f_i(W_0), W_0 \right\rangle \right| \leq \frac{1}{\sqrt{m}} \cdot |S_c| \cdot \frac{4\lambda}{\gamma\sqrt{m}} \leq \frac{16\lambda^2}{\gamma^2\sqrt{m}} + \frac{\lambda\gamma}{4} \leq \frac{\lambda\gamma}{2}, \quad \text{(A.6)}$$

where in the last step we use the condition that $m \geq 4096\lambda^2/\gamma^6$.

- The third term of eq. (A.4) can be bounded as follows: by Lemma 2.3,

$$y_i \left\langle \nabla f_i(W_t), \overline{U} \right\rangle = y_i \left\langle \nabla f_i(W_0), \overline{U} \right\rangle + y_i \left\langle \nabla f_i(W_t) - \nabla f_i(W_0), \overline{U} \right\rangle$$
$$\geq \gamma - \epsilon_1 + y_i \left\langle \nabla f_i(W_t) - \nabla f_i(W_0), \overline{U} \right\rangle.$$

In addition,

$$y_i \left\langle \nabla f_i(W_t) - \nabla f_i(W_0), \overline{U} \right\rangle = y_i \frac{1}{m} \sum_{i=1}^{m} \left( \mathbb{1}\left[ \langle w_{s,t}, x_i \rangle > 0 \right] - \mathbb{1}\left[ \langle w_{s,0}, x_i \rangle > 0 \right] \right) \langle \bar{v}(w_{s,0}), x_i \rangle$$
$$\geq -\frac{1}{m} \cdot |S_c| \geq -\frac{4\lambda}{\gamma\sqrt{m}} - \frac{\epsilon_1}{2} \geq -\frac{\gamma^2}{16} - \frac{\epsilon_1}{2},$$

where we use $m \geq 4096\lambda^2/\gamma^6$. Therefore,

$$y_i \left\langle \nabla f_i(W_t), \overline{U} \right\rangle \geq \gamma - \epsilon_1 - \frac{\gamma^2}{16} - \frac{\epsilon_1}{2} = \gamma - \frac{\gamma^2}{4} \geq \frac{3\gamma}{4}. \quad \text{(A.7)}$$

Putting eqs. (A.5) to (A.7) into eq. (A.4), we have

$$y_i \left\langle \nabla f_i(W_t), \overline{W} \right\rangle \geq -\sqrt{2\ln\left(\frac{4n}{\delta}\right)} - \frac{\lambda\gamma}{2} + \frac{3\lambda\gamma}{4} = \frac{\lambda\gamma}{4} - \sqrt{2\ln\left(\frac{4n}{\delta}\right)} = \ln\left(\frac{4}{\epsilon}\right),$$

for the $\lambda$ given in the statement of Theorem 2.2. Consequently, for any $0 \leq t < t_1$, it holds that $\widehat{\mathcal{R}}^{(t)}\left(\overline{W}\right) \leq \epsilon/4$.

Let $T := \lceil 2\lambda^2/\eta\epsilon \rceil$. The next claim is that $t_1 \geq T$. To see this, note that Lemma 2.6 ensures

$$\left\| W_{t_1} - \overline{W} \right\|_F^2 \leq \left\| W_0 - \overline{W} \right\|_F^2 + 2\eta \left( \sum_{t<t_1} \widehat{\mathcal{R}}^{(t)}\left(\overline{W}\right) \right) \leq \lambda^2 + \frac{\epsilon}{2}\eta t_1.$$

Suppose $t_1 < T$, then we have $t_1 \leq 2\lambda^2/\eta\epsilon$, and thus $\left\| W_{t_1} - \overline{W} \right\|_F^2 \leq 2\lambda^2$. As a result, using $\|\overline{U}\|_F \leq 1$ and the definition of $\overline{W}$,

$$\sqrt{2}\lambda \geq \left\| W_{t_1} - \overline{W} \right\|_F \geq \left\langle W_{t_1} - \overline{W}, \overline{U} \right\rangle = \left\langle W_{t_1} - W_0, \overline{U} \right\rangle - \left\langle \overline{W} - W_0, \overline{U} \right\rangle$$
$$\geq \left\langle W_{t_1} - W_0, \overline{U} \right\rangle - \lambda.$$

Moreover, due to eq. (A.7),

$$\left\langle W_{t_1} - W_0, \overline{U} \right\rangle = -\eta \sum_{\tau<t_1} \left\langle \nabla \widehat{\mathcal{R}}(W_\tau), \overline{U} \right\rangle = \eta \sum_{\tau<t_1} \frac{1}{n} \sum_{i=1}^{n} -\ell'\left(y_i f_i(W_\tau)\right) y_i \left\langle \nabla f_i(W_\tau), \overline{U} \right\rangle$$
$$\geq \eta \sum_{\tau<t_1} \widehat{\mathcal{Q}}(W_\tau) \frac{3\gamma}{4}.$$

As a result,

$$\eta \sum_{\tau < t_1} \widehat{\mathcal{Q}}(W_\tau) \leq \frac{4(\sqrt{2}+1)\lambda}{3\gamma} \leq \frac{4\lambda}{\gamma}.$$

Furthermore, by the triangle inequality, for any $1 \leq s \leq m$

$$
\begin{aligned}
\left\| w_{s,t} - w_{s,0} \right\|_2 &\leq \eta \sum_{\tau < t} \left\| \frac{1}{n} \sum_{i=1}^{n} \ell' \left( y_i f_i(W_\tau) \right) y_i \frac{\partial f_i}{\partial w_{s,\tau}} \right\|_2 \\
&\leq \eta \sum_{\tau < t} \frac{1}{n} \sum_{i=1}^{n} \left| \ell' \left( y_i f_i(W_\tau) \right) \right| \cdot \left\| \frac{\partial f_i}{\partial w_{s,\tau}} \right\|_2 \\
&\leq \eta \sum_{\tau < t} \widehat{\mathcal{Q}}(W_\tau) \frac{1}{\sqrt{m}} \\
&\leq \eta \sum_{\tau < t_1} \widehat{\mathcal{Q}}(W_\tau) \frac{1}{\sqrt{m}} \leq \frac{4\lambda}{\gamma\sqrt{m}},
\end{aligned}
\tag{A.8}
$$

which contradicts the definition of $t_1$. Therefore $t_1 \geq T$.

Now we are ready to prove the claims of Theorem 2.2. The bound on $\left\| w_{s,t} - w_{s,0} \right\|_2$ follow by repeating the steps in eq. (A.8). The risk guarantee follows from Lemma 2.6:

$$\frac{1}{T} \sum_{t<T} \widehat{\mathcal{R}}(W_t) \leq \frac{\left\| W_0 - \overline{W} \right\|_F^2}{\eta T} + \frac{2}{T} \sum_{t<T} \widehat{\mathcal{R}}^{(t)} \left( \overline{W} \right) \leq \frac{\epsilon}{2} + \frac{\epsilon}{2} = \epsilon.$$

$\square$

## B  OMITTED PROOFS FROM SECTION 3

The proof of Theorem 3.2 is based on Rademacher complexity. Given a sample $S = (z_1, \ldots, z_n)$ (where $z_i = (x_i, y_i)$) and a function class $\mathcal{H}$, the Rademacher complexity of $\mathcal{H}$ on $S$ is defined as

$$\mathrm{Rad}\left(\mathcal{H} \circ S\right) := \frac{1}{n} \mathbb{E}_{\epsilon \sim \{-1,+1\}^n} \left[ \sup_{h \in \mathcal{H}} \sum_{i=1}^{n} \epsilon_i h(z_i) \right].$$

We will use the following general result.

**Lemma B.1.** *(Shalev-Shwartz & Ben-David, 2014, Theorem 26.5) If $h(z) \in [a, b]$, then with probability $1 - \delta$,*

$$\sup_{h \in \mathcal{H}} \left( \mathbb{E}_{z \sim \mathcal{D}} \left[ h(z) \right] - \frac{1}{n} \sum_{i=1}^{n} h(z_i) \right) \leq 2\mathrm{Rad}\left(\mathcal{H} \circ S\right) + 3(b-a)\sqrt{\frac{\ln(2/\delta)}{2n}}.$$

We also need the following contraction lemma. Consider a feature sample $X = (x_1, \ldots, x_n)$ and a function class $\mathcal{F}$ on $X$. For each $1 \leq i \leq n$, let $g_i : \mathbb{R} \to \mathbb{R}$ denote a $K$-Lipschitz function. Let $g \circ \mathcal{F}$ denote the class of functions which map $x_i$ to $g_i(f(x_i))$ for some $f \in \mathcal{F}$.

**Lemma B.2.** *(Shalev-Shwartz & Ben-David, 2014, Lemma 26.9)* $\mathrm{Rad}\left(g \circ \mathcal{F} \circ X\right) \leq K\mathrm{Rad}\left(\mathcal{F} \circ X\right)$.

To prove Theorem 3.2, we need one more Rademacher complexity bound. Given a fixed initialization $(W_0, a)$, consider the following classes:

$$\mathcal{W}_\rho := \left\{ W \in \mathbb{R}^{m \times d} \,\middle|\, \left\| w_s - w_{s,0} \right\|_2 \leq \rho \text{ for any } 1 \leq s \leq m \right\},$$

and

$$\mathcal{F}_\rho := \big\{ x \mapsto f(x; W, a) \mid W \in \mathcal{W}_\rho \big\}.$$

Given a feature sample $X$, the following Lemma B.3 controls the Rademacher complexity of $\mathcal{F}_\rho \circ X$. A similar version was given in (Liang, 2016, Theorem 43), and the proof is similar to the proof of (Bartlett & Mendelson, 2002, Theorem 18) which also pushes the supremum through and handles each hidden unit separately.

**Lemma B.3.** $\mathrm{Rad}\big(\mathcal{F}_\rho \circ X\big) \leq \rho\sqrt{m/n}$.

*Proof of Lemma B.3.* We have

$$\mathbb{E}_\epsilon \left[ \sup_{W \in \mathcal{W}_\rho} \sum_{i=1}^{n} \epsilon_i f(x_i; W, a) \right] = \mathbb{E}_\epsilon \left[ \sup_{W \in \mathcal{W}_\rho} \sum_{i=1}^{n} \epsilon_i \sum_{s=1}^{m} \frac{1}{\sqrt{m}} a_s \sigma\big(\langle w_s, x_i \rangle\big) \right]$$

$$= \mathbb{E}_\epsilon \left[ \frac{1}{\sqrt{m}} \sup_{W \in \mathcal{W}_\rho} \sum_{s=1}^{m} \sum_{i=1}^{n} \epsilon_i a_s \sigma\big(\langle w_s, x_i \rangle\big) \right]$$

$$= \mathbb{E}_\epsilon \left[ \frac{1}{\sqrt{m}} \sum_{s=1}^{m} \left( \sup_{\|w_s - w_{s,0}\|_2 \leq \rho} \sum_{i=1}^{n} \epsilon_i a_s \sigma\big(\langle w_s, x_i \rangle\big) \right) \right]$$

$$= \frac{1}{\sqrt{m}} \sum_{i=1}^{m} \mathbb{E}_\epsilon \left[ \sup_{\|w_s - w_{s,0}\|_2 \leq \rho} \sum_{i=1}^{n} \epsilon_i a_s \sigma\big(\langle w_s, x_i \rangle\big) \right].$$

Note that for any $1 \leq s \leq m$, the mapping $z \mapsto a_s \sigma(z)$ is 1-Lipschitz, and thus Lemma B.2 gives

$$\mathbb{E}_\epsilon \left[ \sup_{W \in \mathcal{W}_\rho} \sum_{i=1}^{n} \epsilon_i f(x_i; W, a) \right] \leq \frac{1}{\sqrt{m}} \sum_{i=1}^{m} \mathbb{E}_\epsilon \left[ \sup_{\|w_s - w_{s,0}\|_2 \leq \rho} \sum_{i=1}^{n} \epsilon_i a_s \sigma\big(\langle w_s, x_i \rangle\big) \right]$$

$$\leq \frac{1}{\sqrt{m}} \sum_{i=1}^{m} \mathbb{E}_\epsilon \left[ \sup_{\|w_s - w_{s,0}\|_2 \leq \rho} \sum_{i=1}^{n} \epsilon_i \langle w_s, x_i \rangle \right].$$

Invoking the Rademacher complexity of linear classifiers (Shalev-Shwartz & Ben-David, 2014, Lemma 26.10) then gives

$$\mathrm{Rad}\big(\mathcal{F}_\rho \circ X\big) = \frac{1}{n} \mathbb{E}_\epsilon \left[ \sup_{W \in \mathcal{W}_\rho} \sum_{i=1}^{n} \epsilon_i f(x_i; W, a) \right] \leq \frac{\rho\sqrt{m}}{\sqrt{n}}.$$

$\square$

Now we are ready to prove the main generalization result Theorem 3.2.

*Proof.* Fix an initialization $(W_0, a)$, and let $\mathcal{H} := \Big\{ (x, y) \mapsto -\ell'\big(yf(x)\big) \mid f \in \mathcal{F}_\rho \Big\}$. Since for any $h \in \mathcal{H}$ and any $z$, $h(z) \in [0, 1]$, Lemma B.1 ensures that with probability $1 - \delta$ over the data sampling,

$$\sup_{h \in \mathcal{H}} \left( \mathbb{E}_{z \sim \mathcal{D}}\big[h(z)\big] - \frac{1}{n} \sum_{i=1}^{n} h(z_i) \right) = \sup_{W \in \mathcal{W}_\rho} \Big( \mathcal{Q}(W) - \widehat{\mathcal{Q}}(W) \Big) \leq 2\mathrm{Rad}\big(\mathcal{H} \circ S\big) + 3\sqrt{\frac{\ln(2/\delta)}{2n}}.$$

Since for each $1 \leq i \leq n$, the mapping $z \mapsto -\ell'(y_i z)$ is $(1/4)$-Lipschitz, Lemma B.2 further ensures that $\mathrm{Rad}\big(\mathcal{H} \circ S\big) \leq \mathrm{Rad}\big(\mathcal{F}_\rho \circ X\big)/4$, and thus

$$\sup_{W \in \mathcal{W}_\rho} \Big( \mathcal{Q}(W) - \widehat{\mathcal{Q}}(W) \Big) \leq \frac{\rho\sqrt{m}}{2\sqrt{n}} + 3\sqrt{\frac{\ln(2/\delta)}{2n}}. \tag{B.1}$$

On the other hand, Theorem 2.2 ensures that under the conditions of Theorem 3.2, for any fixed dataset, with probability $1 - 3\delta$ over the random initialization, we have

$$\widehat{\mathcal{Q}}(W_k) \leq \widehat{\mathcal{R}}(W_k) \leq \epsilon, \quad \text{and} \quad \left\|w_{s,k} - w_{s,0}\right\|_2 \leq \frac{4\lambda}{\gamma\sqrt{m}}.$$

As a result, invoking eq. (B.1) with $\rho = 4\lambda/(\gamma\sqrt{m})$, with probability $1 - 4\delta$ over the random initialization and data sampling,

$$\mathcal{Q}(W_k) \leq \widehat{\mathcal{Q}}(W_k) + \frac{2\lambda}{\gamma\sqrt{n}} + 3\sqrt{\frac{\ln(2/\delta)}{2n}} \leq \epsilon + \frac{8\left(\sqrt{2\ln(4n/\delta)} + \ln(4/\epsilon)\right)}{\gamma^2\sqrt{n}} + 3\sqrt{\frac{\ln(2/\delta)}{2n}}.$$

Invoking $P_{(x,y)\sim\mathcal{D}}\left(yf(x; W, a) \leq 0\right) \leq 2\mathcal{Q}(W)$ finishes the proof. □

## C   OMITTED PROOFS FROM SECTION 4

*Proof of Lemma 4.2.* Recall that $\left\|\nabla f_t(W_t)\right\|_F \leq 1$, we have

$$\left\|W_{t+1} - \overline{W}\right\|_F^2 \leq \left\|W_t - \overline{W}\right\|_F^2 - 2\eta\ell'\left(y_t f_t(W_t)\right)y_t\left\langle\nabla f_t(W_t), W_t - \overline{W}\right\rangle + \eta^2\left(\ell'\left(y_t f_t(W_t)\right)\right)^2.$$
$$(C.1)$$

Similar to the proof of Lemma 2.6, the first order term of eq. (C.1) can be handled using the convexity of $\ell$ and homogeneity of ReLU as follows

$$\ell'\left(y_t f_t(W_t)\right)y_t\left\langle\nabla f_t(W_t), W_t - \overline{W}\right\rangle \geq \mathcal{R}_t(W_t) - \mathcal{R}_t\left(\overline{W}\right), \quad (C.2)$$

and the second-order term of eq. (C.1) can be bounded as follows

$$\eta^2\left(\ell'\left(y_t f_t(W_t)\right)\right)^2 \leq -\eta\ell'\left(y_t f_t(W_t)\right) \leq \eta\ell\left(y_t f_t(W_t)\right) = \eta\mathcal{R}_t(W_t), \quad (C.3)$$

since $\eta, -\ell' \leq 1$ and $-\ell' \leq \ell$. Combining eqs. (C.1) to (C.3) gives

$$\eta\mathcal{R}_t(W_t) \leq \left\|W_t - \overline{W}\right\|_F^2 - \left\|W_{t+1} - \overline{W}\right\|_F^2 + 2\eta\mathcal{R}_t\left(\overline{W}\right).$$

Telescoping gives the claim. □

With Lemma 4.2, we give the following result, which is an extension of Theorem 2.2 to the SGD setting.

**Lemma C.1.** *Under Assumption 3.1, given any $\epsilon \in (0,1)$, any $\delta \in (0, 1/3)$, and any positive integer $n_0$, let*

$$\lambda := \frac{\sqrt{2\ln(4n_0/\delta)} + \ln(4/\epsilon)}{\gamma/4}, \quad \text{and} \quad M := \frac{4096\lambda^2}{\gamma^6}.$$

*For any $m \geq M$ and any constant step size $\eta \leq 1$, if $n_0 \geq n := \lceil 2\lambda^2/\eta\epsilon \rceil$, then with probability $1 - 3\delta$,*

$$\frac{1}{n}\sum_{i<n}\mathcal{Q}_i(W_i) \leq \epsilon.$$

*Proof.* We first sample $n_0$ data examples $(x_0, y_0), \ldots, (x_{n_0-1}, y_{n_0-1})$, and then feed $(x_i, y_i)$ to SGD at step $i$. We only consider the first $n_0$ steps.

The proof is similar to the proof of Theorem 2.2. Let $n_1$ denote the first step before $n_0$ such that there exists some $1 \leq s \leq m$ with $\left\|w_{s,n_1} - w_{s,0}\right\|_2 > 4\lambda/(\gamma\sqrt{m})$. If such a step does not exist, let $n_1 = n_0$.

Let $\overline{W} := W_0 + \lambda\overline{U}$, in exactly the same way as in Theorem 2.2, we can show that with probability $1 - 3\delta$, for any $0 \leq i < n_1$,

$$y_i\left\langle\nabla f_i(W_i), \overline{W}\right\rangle \geq \ln\left(\frac{4}{\epsilon}\right), \quad \text{and thus} \quad \mathcal{R}_i\left(\overline{W}\right) \leq \epsilon/4.$$

Now consider $n := \lceil 2\lambda^2/\eta\epsilon \rceil$. Using Lemma 4.2, in the same way as the proof of Theorem 2.2 (replacing $\widehat{\mathcal{Q}}(W_\tau)$ with $\mathcal{Q}_i(W_i)$, etc.), we can show that $n \leq n_1$. Then invoking Lemma 4.2 again, we get

$$\frac{1}{n}\sum_{i<n}\mathcal{Q}_i(W_i) \leq \frac{1}{n}\sum_{i<n}\mathcal{R}_i(W_i) \leq \frac{\left\|W_0 - \overline{W}\right\|_F^2}{\eta n} + \frac{2}{n}\sum_{i<n}\mathcal{R}_i\left(\overline{W}\right) \leq \frac{\epsilon}{2} + \frac{\epsilon}{2} = \epsilon.$$

$\square$

Next we prove Lemma 4.3. We need the following martingale Bernstein bound.

**Lemma C.2.** *(Beygelzimer et al., 2011, Theorem 1) Let $(M_t, \mathcal{F}_t)_{t\geq 0}$ denote a martingale with $M_0 = 0$ and $\mathcal{F}_0$ be the trivial $\sigma$-algebra. Let $(\Delta_t)_{t\geq 1}$ denote the corresponding martingale difference sequence, and let*

$$V_t := \sum_{j=1}^t \mathbb{E}\left[\Delta_j^2 \Big| \mathcal{F}_{j-1}\right]$$

*denote the sequence of conditional variance. If $\Delta_t \leq R$ a.s., then for any $\delta \in (0,1)$, with probability at least $1 - \delta$,*

$$M_t \leq \frac{V_t}{R}(e-2) + R\ln\left(\frac{1}{\delta}\right).$$

*Proof of Lemma 4.3.* For any $i \geq 0$, let $z_i$ denote $(x_i, y_i)$, and $z_{0,i}$ denote $(z_0, \ldots, z_i)$. Note that the quantity $\sum_{t<i}\left(\mathcal{Q}(W_t) - \mathcal{Q}_t(W_t)\right)$ is a martingale w.r.t. the filtration $\sigma(z_{0,i-1})$. The martingale difference sequence is given by $\mathcal{Q}(W_t) - \mathcal{Q}_t(W_t)$, which satisfies

$$\mathcal{Q}(W_t) - \mathcal{Q}_t(W_t) = \mathbb{E}_{(x,y)\sim\mathcal{D}}\left[-\ell'\left(yf(x;W_t,a)\right)\right] + \ell'\left(y_t f(x_t;W_t,a)\right) \leq 1, \qquad \text{(C.4)}$$

since $-1 \leq \ell' \leq 0$. Moreover, we have

$$
\begin{aligned}
&\mathbb{E}\left[\left(\mathcal{Q}(W_t) - \mathcal{Q}_t(W_t)\right)^2 \Big| \sigma(z_{0,t-1})\right] \\
&= \mathcal{Q}(W_t)^2 - 2\mathcal{Q}(W_t)\mathbb{E}\left[\mathcal{Q}_t(W_t)\big|\sigma(z_{0,t-1})\right] + \mathbb{E}\left[\mathcal{Q}_t(W_t)^2\big|\sigma(z_{0,t-1})\right] \\
&= -\mathcal{Q}(W_t)^2 + \mathbb{E}\left[\mathcal{Q}_t(W_t)^2\big|\sigma(z_{0,t-1})\right] \\
&\leq \mathbb{E}\left[\mathcal{Q}_t(W_t)^2\big|\sigma(z_{0,t-1})\right] \\
&\leq \mathbb{E}\left[\mathcal{Q}_t(W_t)\big|\sigma(z_{0,t-1})\right] \\
&= \mathcal{Q}(W_t).
\end{aligned}
\qquad \text{(C.5)}
$$

Invoking Lemma C.2 with eqs. (C.4) and (C.5) gives that with probability $1 - \delta$,

$$\sum_{t<i}\left(\mathcal{Q}(W_t) - \mathcal{Q}_t(W_t)\right) \leq (e-2)\sum_{t<i}\mathcal{Q}(W_t) + \ln\left(\frac{1}{\delta}\right).$$

Consequently,

$$\sum_{t<i}\mathcal{Q}(W_t) \leq 4\sum_{t<i}\mathcal{Q}_t(W_t) + 4\ln\left(\frac{1}{\delta}\right).$$

$\square$

Finally, we prove Theorem 4.1.

*Proof of Theorem 4.1.* Suppose the condition of Lemma C.1 holds. Then we have for $n = \lceil 2\lambda^2/\eta\epsilon \rceil$, with probability $1 - 3\delta$,

$$\frac{1}{n}\sum_{i<n}\mathcal{Q}_i(W_i) \leq \epsilon.$$

Further invoking Lemma 4.3 gives that with probability $1 - 4\delta$,

$$\frac{1}{n}\sum_{i<n}\mathcal{Q}(W_i) \leq \frac{4}{n}\sum_{i<n}\mathcal{Q}_i(W_i) + \frac{4}{n}\ln\left(\frac{1}{\delta}\right) \leq 5\epsilon.$$

Since $P_{(x,y)\sim\mathcal{D}}\left(yf(x;W,a) \leq 0\right) \leq 2\mathcal{Q}(W)$, we get

$$\frac{1}{n}\sum_{i=1}^{n}P_{(x,y)\sim\mathcal{D}}\left(yf(x;W_i,a) \leq 0\right) \leq 10\epsilon.$$

For the condition of Lemma C.1 to hold, it is enough to let

$$n_0 = \Theta\left(\frac{\ln(1/\delta)}{\eta\gamma^2\epsilon^2}\right),$$

which gives

$$M = \Theta\left(\frac{\ln(1/\delta) + \ln(1/\epsilon)^2}{\gamma^8}\right) \quad \text{and} \quad n = \Theta\left(\frac{\ln(1/\delta) + \ln(1/\epsilon)^2}{\gamma^2\epsilon}\right).$$

$\square$

## D OMITTED PROOFS FROM SECTION 5

*Proof of Proposition 5.1.* Define $f : \mathcal{H} \to \mathbb{R}$ by

$$f(w) := \frac{1}{2}\int \|w(z)\|_2^2 \, \mathrm{d}\mu_{\mathcal{N}}(z) = \frac{1}{2}\|w\|_{\mathcal{H}}^2.$$

It holds that $f$ is continuous, and $f^*$ has the same form. Define $g : \mathbb{R}^n \to \mathbb{R}$ by

$$g(p) := \max_{1\leq i\leq n} p_i,$$

with conjugate

$$g^*(q) = \begin{cases} 0, & \text{if } q \in \Delta_n, \\ +\infty, & \text{o.w.} \end{cases}$$

Finally, define the linear mapping $A : \mathcal{H} \to \mathbb{R}^n$ by $(Aw)_i = y_i \langle w, \phi_i\rangle_{\mathcal{H}}$.

Since $f$, $f^*$, $g$ and $g^*$ are lower semi-continuous, and $\mathbf{dom}\,g - A\mathbf{dom}\,f = \mathbb{R}^n$, and $\mathbf{dom}\,f^* - A^*\mathbf{dom}\,g^* = \mathcal{H}$, Fenchel duality may be applied in each direction (Borwein & Zhu, 2005, Theorem 4.4.3), and ensures that

$$\inf_{w\in\mathcal{H}}\left(f(w) + g(Aw)\right) = \sup_{q\in\mathbb{R}^n}\left(-f^*(A^*q) - g^*(-q)\right).$$

with optimal primal-dual solutions $(\bar{w}, \bar{q})$. Moreover

$$\begin{aligned}
\inf_{w\in\mathcal{H}}\left(f(w) + g(Aw)\right) &= \inf_{w\in\mathcal{H},u\in\mathbb{R}^n}\sup_{q\in\mathbb{R}^n}\left(f(w) + g(Aw+u) + \langle q, u\rangle\right) \\
&\geq \sup_{q\in\mathbb{R}^n}\inf_{w\in\mathcal{H},u\in\mathbb{R}^n}\left(f(w) + g(Aw+u) + \langle q, u\rangle\right) \\
&= \sup_{q\in\mathbb{R}^n}\inf_{w\in\mathcal{H},u\in\mathbb{R}^n}\left(\left(f(w) - \langle A^*q, w\rangle\right)_{\mathcal{H}} + \left(g(Aw+u) - \langle -q, Aw+u\rangle\right)\right) \\
&= \sup_{q\in\mathbb{R}^n}\left(-f^*(A^*q) - g^*(-q)\right).
\end{aligned}$$

By strong duality, the inequality holds with equality. It follows that

$$\bar{w} = A^*\bar{q}, \quad \text{and} \quad \mathbf{supp}(-\bar{q}) \subset \operatorname*{arg\,max}_{1\leq i\leq n}(A\bar{w})_i.$$

Now let us look at the dual optimization problem. It is clear that

$$\sup_{q \in \mathbb{R}^n} \left( -f^*(A^*q) - g^*(-q) \right) = -\inf_{q \in \Delta_n} f^*(A^*q).$$

In addition, we have

$$f^*(A^*q) = \frac{1}{2} \int \left\| \sum_{i=1}^n q_i y_i \phi_i(z) \right\|_2^2 \mathrm{d}\mu_\mathcal{N}(z)$$

$$= \frac{1}{2} \int \sum_{i,j=1}^n q_i q_j y_i y_j \langle \phi_i(z), \phi_j(z) \rangle \mathrm{d}\mu_\mathcal{N}(z)$$

$$= \frac{1}{2} \sum_{i,j=1}^n q_i q_j y_i y_j \int \langle \phi_i(z), \phi_j(z) \rangle \mathrm{d}\mu_\mathcal{N}(z)$$

$$= \frac{1}{2} \sum_{i,j=1}^n q_i q_j y_i y_j K_1(i,j) = \frac{1}{2}(q \odot y)^\top K_1(q \odot y),$$

and thus $f^*(A^*\bar{q}) = \gamma_1^2/2$. Since $\bar{w} = A^*\bar{q}$, we have that $\|\bar{w}\|_\mathcal{H} = \gamma_1$. In addition,

$$g(A\bar{w}) = -f^*\left(A^*\bar{q}\right) - f\left(\bar{w}\right) = -\gamma_1^2,$$

and thus $-\bar{w}$ has margin $\gamma_1^2$. Moreover, we have

$$\bar{w}(z) = \sum_{i=1}^n \bar{q}_i y_i \phi_i(z) = \sum_{i=1}^n \bar{q}_i y_i x_i \mathbb{1}\left[\langle z, x_i \rangle > 0\right],$$

and thus $\|\bar{w}(z)\|_2 \le 1$. Therefore, $\hat{v} = -\bar{w}/\gamma_1$ satisfies all requirements of Proposition 5.1. $\qquad \square$

*Proof of Proposition 5.2.* Let $\hat{q}$ denote the uniform probability vector $(1/n, \ldots, 1/n)$. Note that

$$\mathbb{E}_{\epsilon \sim \mathrm{unif}\left(\{-1,+1\}^n\right)} \left[(\hat{q} \odot \epsilon)^\top K_1 (\hat{q} \odot \epsilon)\right] = \mathbb{E}_{\epsilon \sim \mathrm{unif}\left(\{-1,+1\}^n\right)} \left[\sum_{i,j=1}^n \frac{1}{n^2} \epsilon_i \epsilon_j K_1(x_i, x_j)\right]$$

$$= \frac{1}{n^2} \sum_{i,j=1}^n \mathbb{E}_{\epsilon \sim \mathrm{unif}\left(\{-1,+1\}^n\right)} \left[\epsilon_i \epsilon_j K_1(x_i, x_j)\right]$$

$$= \frac{1}{n^2} \sum_{i=1}^n K_1(x_i, x_i) = \frac{1}{2n}.$$

Since $0 \le (\hat{q} \odot \epsilon)^\top K_1 (\hat{q} \odot \epsilon) \le 1$ for any $\epsilon$, by Markov's inequality with probability 0.9, it holds that $(\hat{q} \odot \epsilon)^\top K_1 (\hat{q} \odot \epsilon) \le 1/(20n)$, and thus $\gamma_1 \le 1/\sqrt{20n}$. $\qquad \square$

*Proof of Proposition 5.3.* By symmetry, we only need to consider an $(x, y)$ where $(x_1, x_2, y) = (1/\sqrt{d-1}, 0, 1)$. Let $z_{p,q}$ denote $(z_p, z_{p+1}, \ldots, z_q)$, and similarly define $x_{p,q}$. We have

$$y \int \langle \bar{v}(z), x \rangle \mathbb{1}\left[\langle z, x \rangle > 0\right] \mathrm{d}\mu_\mathcal{N}(z)$$

$$= y \int \left( \int \langle \bar{v}(z), x \rangle \mathbb{1}\left[\langle z, x \rangle > 0\right] \mathrm{d}\mu_\mathcal{N}(z_{3,d}) \right) \mathrm{d}\mu_\mathcal{N}(z_{1,2}) \tag{D.1}$$

$$= y \int \langle \bar{v}(z)_{1,2}, x_{1,2} \rangle \left( \int \mathbb{1}\left[\langle z_{1,2}, x_{1,2} \rangle + \langle z_{3,d}, x_{3,d} \rangle > 0\right] \mathrm{d}\mu_\mathcal{N}(z_{3,d}) \right) \mathrm{d}\mu_\mathcal{N}(z_{1,2}) \tag{D.2}$$

$$= \sum_{i=1}^4 y \int \langle \bar{v}(z)_{1,2}, x_{1,2} \rangle \left( \int \mathbb{1}\left[\langle z_{1,2}, x_{1,2} \rangle + \langle z_{3,d}, x_{3,d} \rangle > 0\right] \mathrm{d}\mu_\mathcal{N}(z_{3,d}) \right) \mathbb{1}\left[z_{1,2} \in A_i\right] \mathrm{d}\mu_\mathcal{N}(z_{1,2}),$$

$$\tag{D.3}$$

where eq. (D.1) is due to the independence between $z_{1,2}$ and $z_{3,d}$, and in eq. (D.2) we use the fact that $\bar{v}(z)_{1,2}$ only depends on $z_{1,2}$ and $\bar{v}(z)_{3,d}$ are all zero. Since $\langle \bar{v}(z)_{1,2}, x_{1,2} \rangle = 0$ for $z_{1,2} \in A_2 \cup A_4$, we only need to consider $A_1$ and $A_3$ in eq. (D.3). For simplicity, we will denote $z_{1,2}$ by $p \in \mathbb{R}^2$, and $\bar{v}(z)_{1,2}$ by $\bar{v}(p)$, and $z_{3,d}$ by $q \in \mathbb{R}^{d-2}$.

For any nonzero $p \in A_1$, we have $-p \in A_3$, and $\langle \bar{v}(p), x_{1,2} \rangle = 1/\sqrt{d-1}$. Therefore

$$
\begin{aligned}
y \langle \bar{v}(p), x_{1,2} \rangle &\left( \int \mathbb{1} \left[ \langle p, x_{1,2} \rangle + \langle q, x_{3,d} \rangle > 0 \right] \mathrm{d}\mu_{\mathcal{N}}(q) \right) \\
&+ y \langle \bar{v}(-p), x_{1,2} \rangle \left( \int \mathbb{1} \left[ \langle -p, x_{1,2} \rangle + \langle q, x_{3,d} \rangle > 0 \right] \mathrm{d}\mu_{\mathcal{N}}(q) \right) \\
&= \frac{1}{\sqrt{d-1}} \int \left( \mathbb{1} \left[ \frac{p_1}{\sqrt{d-1}} + \langle q, x_{3,d} \rangle > 0 \right] - \mathbb{1} \left[ \frac{-p_1}{\sqrt{d-1}} + \langle q, x_{3,d} \rangle > 0 \right] \right) \mathrm{d}\mu_{\mathcal{N}}(q) \\
&= \frac{1}{\sqrt{d-1}} \mathbb{P} \left( \frac{-p_1}{\sqrt{d-1}} \leq \langle q, x_{3,d} \rangle \leq \frac{p_1}{\sqrt{d-1}} \right).
\end{aligned}
\tag{D.4}
$$

Let $\varphi$ denote the density function of the standard Gaussian distribution, and for $c > 0$, let $U(c)$ denote the probability that a standard Gaussian random variable lies in the interval $[-c, c]$:

$$
U(c) := \int_{-c}^{c} \varphi(t) \, \mathrm{d}t.
$$

Since $\langle q, x_{3,d} \rangle$ is a Gaussian variable with standard deviation $\sqrt{(d-2)/(d-1)}$, we have

$$
\mathbb{P} \left( \frac{-p_1}{\sqrt{d-1}} \leq \langle q, x_{3,d} \rangle \leq \frac{p_1}{\sqrt{d-1}} \right) = U \left( \frac{p_1}{\sqrt{d-2}} \right).
\tag{D.5}
$$

Plugging eqs. (D.4) and (D.5) into eq. (D.3) gives:

$$
\begin{aligned}
y \int \langle \bar{v}(z), x \rangle \mathbb{1} \left[ \langle z, x \rangle > 0 \right] \mathrm{d}\mu_{\mathcal{N}}(z) &= \frac{1}{\sqrt{d-1}} \int U \left( \frac{p_1}{\sqrt{d-2}} \right) \mathbb{1} \left[ p \in A_1 \right] \mathrm{d}\mu_{\mathcal{N}}(p) \\
&= \frac{1}{\sqrt{d-1}} \int_0^\infty U \left( \frac{p_1}{\sqrt{d-2}} \right) \left( \int_{-p_1}^{p_1} \varphi(p_2) \, \mathrm{d}p_2 \right) \varphi(p_1) \, \mathrm{d}p_1 \\
&= \frac{1}{\sqrt{d-1}} \int_0^\infty U \left( \frac{p_1}{\sqrt{d-2}} \right) U(p_1) \varphi(p_1) \, \mathrm{d}p_1 \\
&\geq \frac{1}{\sqrt{d-1}} \int_0^1 U \left( \frac{p_1}{\sqrt{d-2}} \right) U(p_1) \varphi(p_1) \, \mathrm{d}p_1.
\end{aligned}
$$

For $t \in [-1, +1]$, it holds that $\varphi(t) \geq 1\sqrt{2\pi e}$, and thus

$$
U(a) = \int_{-a}^{a} \varphi(t) \, \mathrm{d}t \geq \frac{2a}{\sqrt{2\pi e}}.
$$

Therefore eq. (D.3) is lower bounded by

$$
\begin{aligned}
\frac{1}{\sqrt{d-1}} \int_0^1 U \left( \frac{p_1}{\sqrt{d-2}} \right) U(p_1) \varphi(p_1) \, \mathrm{d}p_1 &\geq \frac{1}{\sqrt{d-1}} \int_0^1 \frac{2}{\sqrt{2\pi e}} \cdot \frac{p_1}{\sqrt{d-2}} \cdot \frac{2p_1}{\sqrt{2\pi e}} \cdot \frac{1}{\sqrt{2\pi e}} \, \mathrm{d}p_1 \\
&\geq \frac{1}{20\sqrt{(d-1)(d-2)}} \int_0^1 p_1^2 \, \mathrm{d}p_1 \\
&= \frac{1}{60\sqrt{(d-1)(d-2)}} \\
&\geq \frac{1}{60d}.
\end{aligned}
$$

$\square$

To prove Proposition 5.4, we need the following technical lemma.

**Lemma D.1.** *Given $z_1 \sim \mathcal{N}(0,1)$ and $z_2 \sim \mathcal{N}(0, b^2)$ that are independent where $b > 1$, we have*

$$\mathbb{P}\left(|z_1| < |z_2|\right) > 1 - \frac{1}{b}.$$

*Proof.* First note that for $z_3 \sim \mathcal{N}(0,1)$ which is independent of $z_1$,

$$\mathbb{P}\left(|z_1| < |z_2|\right) = \mathbb{P}\left(|z_1| < b|z_3|\right) = 1 - \mathbb{P}\left(|z_3| < \frac{1}{b}|z_1|\right).$$

Still let $\varphi$ denote the density of $\mathcal{N}(0,1)$, and let $U(c)$ denote the probability that $z_3 \in [-c, c]$. We have

$$\mathbb{P}\left(|z_3| < \frac{1}{b}|z_1|\right) = \int \int \mathbb{1}\left[|z_3| < \frac{1}{b}|z_1|\right] \varphi(z_3)\varphi(z_1)\,\mathrm{d}z_3\,\mathrm{d}z_1$$

$$= \int U\left(\frac{1}{b}|z_1|\right)\varphi(z_1)\,\mathrm{d}z_1$$

$$\leq \frac{2}{\sqrt{2\pi}b} \int |z_1|\varphi(z_1)\,\mathrm{d}z_1 = \frac{2}{\pi b} < \frac{1}{b},$$

where we use the facts that $U(c) \leq 2c/\sqrt{2\pi}$ and $\mathbb{E}[|z_1|] = \sqrt{2/\pi}$. $\qquad\square$

We now give the proof of Proposition 5.4 using Lemma D.1.

*Proof of Proposition 5.4.* By symmetry, we only need to consider the following training set:

$$\begin{aligned}
x_1 &= (1, 0, 1, \ldots, 1), & y_1 &= 1, \\
x_2 &= (0, 1, 1, \ldots, 1), & y_2 &= -1, \\
x_3 &= (-1, 0, 1, \ldots, 1), & y_3 &= 1, \\
x_4 &= (0, -1, 1, \ldots, 1), & y_4 &= -1.
\end{aligned}$$

The $1/\sqrt{d-1}$ factor is omitted also because we only discuss the $0/1$ loss.

For any $s$, let $A_s$ denote the event that

$$\mathbb{1}\left[\langle w_s, x_1 \rangle > 0\right] = \mathbb{1}\left[\langle w_s, x_2 \rangle > 0\right] = \mathbb{1}\left[\langle w_s, x_3 \rangle > 0\right] = \mathbb{1}\left[\langle w_s, x_4 \rangle > 0\right].$$

We will show that if $m \leq \sqrt{d-2}/4$, then $A_s$ is true for all $1 \leq s \leq m$ with probability $1/2$, and Proposition 5.4 follows from the fact that the XOR data is not linearly separable.

For any $s$ and $i$,

$$\langle w_s, x_i \rangle = (w_s)_1 (x_i)_1 + (w_s)_2 (x_i)_2 + \sum_{j=3}^{d} (w_s)_j.$$

Since $\left((x_i)_1, (x_i)_2\right)$ is $(1, 0)$ or $(0, 1)$ or $(-1, 0)$ or $(0, -1)$, event $A_s$ will happen as long as

$$\left|(w_s)_1\right| < \left|\sum_{j=3}^{d} (w_s)_j\right|, \quad \text{and} \quad \left|(w_s)_2\right| < \left|\sum_{s=3}^{d} (w_s)_j\right|.$$

Note that $(w_s)_1, (w_s)_2 \sim \mathcal{N}(0,1)$ while $\sum_{j=3}^{d} (w_s)_j \sim \mathcal{N}(0, d-2)$. As a result, due to Lemma D.1,

$$\mathbb{P}\left(\left|(w_s)_1\right| < \left|\sum_{j=3}^{d} (w_s)_j\right|\right) = \mathbb{P}\left(\left|(w_s)_2\right| < \left|\sum_{s=3}^{d} (w_s)_j\right|\right) > 1 - \frac{1}{\sqrt{d-2}}.$$

Using a union bound, $\mathbb{P}(A_s) > 1 - 2/\sqrt{d-2}$. If $m \leq \sqrt{d-2}/4$, then by a union bound again,

$$\mathbb{P}\left(\bigcup_{1 \leq s \leq m} A_s\right) > 1 - \frac{2}{\sqrt{d-2}}m \geq 1 - \frac{2}{\sqrt{d-2}}\frac{\sqrt{d-2}}{4} = \frac{1}{2}.$$

$\qquad\square$

