# OpenReview forum: "Polylogarithmic width suffices for gradient descent to achieve arbitrarily small test error with shallow ReLU networks"
_ICLR.cc/2020/Conference — Accept (Poster)_

### Official Review · AnonReviewer1 · 2019-10-14
**Official Blind Review #1**

**Rating:** 8

**Review:**

Summary and Decision

The authors of this paper studied the optimization and generalization properties of shallow ReLU networks. In particular, the authors were able to show a width dependence that is polylogarithmic on the number of samples, probability or failure, error tolerance, and a margin parameter. This work is unique in that the authors showed how to bound many key quantities in terms of the margin parameter, and drew a connection with the neural tangent kernel's maximum margin. Furthermore, the overall reading experience was very smooth, although I do have some minor comments later.

The main concern from me is on the implicit dependence of the margin parameter \gamma, most notably this can lead to the width dependence to be polynomial in terms of the number of samples and the minimum separation distance. While this concern warrants a careful discussion (below), I believe the paper still offers a nice analysis of shallow networks.

Overall, I would recommend accept for this paper.


Background

There has been a large number of works studying very wide networks, showing both optimization and generalization results. While there has been great progress, most of these existing results require the width of both deep (and shallow) networks to be very large. For example, even by being polynomial in the number of samples, the networks are already unrealizable in practice. Therefore, guarantees with much better dependence is highly desirable.


Discussion of Contributions

As the title may suggest, it is a bit surprising that we can show that polylog width is sufficient. Intuitively, we can imagine that the classification margin can grow exponentially more complex as the number of samples increase. I believe the nice result can be attributed to a careful analysis of key quantities such as \| W_t - W_0 \|_F in terms of the margin parameter from Assumption 2.1.

Some nice examples of the analysis in this paper include the introduction of the weight matrix \overline{U} and \overline{W} as an intermediate between W_0 and W_t, and observing that the activations of the ReLU \xi_{i,t} do not change very much during training. The tricks together led to a very tight bound on the change in weights \| W_t - W_0 \|_F in terms of the margin parameter \gamma. As the authors mentioned on page 6, this tight control was used to bound the Rademacher complexity later.

The connection drawn between the margin assumption and neural tangent kernel (Proposition 5.1) is also interesting on its own. The authors intended this result to serve as a justification of the margin assumption (2.1).


Discussion of the Margin Parameter

Let me start by saying I'm not completely certain on how to interpret this margin parameter \gamma in Assumption 2.1. Perhaps I'm missing some obvious ideas here, but I would still like the authors respond with some more details. At the same time, I don't believe this is a sufficient criticism to reject this paper, as I believe the analysis in terms of \gamma is still valuable.

On one hand, if we were to assume the margin condition holds for all possible data points (i.e. Assumption 3.1), then there is no concern about polynomial dependence on the number of samples, and this is certainly a reasonable assumption in some applications.

On the other hand, many of the previous analysis on wide networks were in terms of a minimum separation distance, i.e. assume there exists a \delta > 0 such that for all i \neq j, we have
	\| x_i - x_j \| \geq \delta .
The authors have provided a discussion in section 5, including both a worst case bound of
	\gamma \geq \delta / (100 n^2),
by Oymak and Soltanokotabi (2019) and an example where the margin is O( n^{-1/2} ) with high probability.

Using either bounds on \gamma, we will have a width with polynomial dependence in terms of the number of samples and minimum separation distance. Therefore if we were to compare against previous works in the same benchmark, i.e. using a minimum separation assumption instead, then arguably this work did not achieve a width that is only polylog in terms of the number of samples.

That being said, I don't believe the authors were intentionally trying to hide sample dependence inside an assumption. The paper is presented in a very transparent way, and the authors were being honest in chapter 5 about the worst case dependence on the number of samples.

To summarize, it is unclear to me whether the paper truly achieved a width dependence that is polylog in terms of the number of samples, but the analysis in terms of \gamma remains a valuable contribution. I welcome the authors and the other reviewers to provide additional comments on whether the title and claims of this paper is appropriate.


Minor Comments

For the sake of improving readability, I also have some minor comments that do not contribute towards the decision.

 - On page 5, the first observation bullet point in section 2.2, it is written here that by triangle inequality
     \| \nabla \widehat{R}(W) \|_F \leq \widehat{Q}(W) ,
    I thought you needed an absolute value on the right hand side, perhaps you should mention that \ell is strictly decreasing.

 - On page 6, in the statement of Corollary 3.3, you are missing \eta \leq 1, and perhaps the \tilde on \Omega and O should be defined.
 - On the same note, while it is obvious that Theorem 3.2 implies this corollary, it is still worth writing up a proof to compute the constants.

 - On page 7, the notation \Delta_n and \odot are not defined, I had to infer definition from the proof.

 - On page 12, just below the second equation, it wasn't immediately clear how \widehat{R}( \overline{W} ) \leq \epsilon / 4. I believe it's worth expanding the definitions a bit, and explicitly plug in Lemma 2.5.

 - On page 12, in the second last equation, I'm actually not sure where this inequality comes from
      \| W_t - \overline{W} \|_F \geq \langle W_t - \overline {W} , \overline{U} \rangle
    Perhaps it's obvious, but I currently don't see it.


**Experience Assessment:**

I have read many papers in this area.

**Review Assessment: Checking Correctness Of Derivations And Theory:**

I carefully checked the derivations and theory.

**Review Assessment: Checking Correctness Of Experiments:**

N/A

**Review Assessment: Thoroughness In Paper Reading:**

I read the paper thoroughly.

---

> ### Author Response · Authors · 2019-11-11
> **Response to AnonReviewer #1**
>
> We thank the reviewer for their review and support.
>
> First we want to note that as AnonReviewer #2 points out, there was a term not handled in the original analysis. We have fixed this bug in the revision. The missing term is small for reasons common to NTK analyses, namely that weights stay close to initialization; other parts of the proof remain unchanged. The required width still has only a polylogarithmic dependency on n, 1/delta, and 1/epsilon, but now it depends on 1/gamma^8.
>
> Next we discuss the separability assumption. Let us mention that the Hilbert space \mathcal{H} defined in section 2 is just the RKHS induced by the NTK, and the training feature x_i is mapped to \phi_i which lies in the RKHS. The margin gamma given by Assumption 2.1 and 3.1 is just the separation margin in the RKHS. More discussion has been added to the beginning of Section 2 and 5 of the revision.
>
> In Section 5 we give various estimates of gamma which depends on poly(n). However, these bounds are for arbitrary labels or random labels. In general it might be impossible to prove a polylogarithmic width for arbitrary or random labels, and this might also explain the poly(n) dependency in prior work, since their bounds usually hold for arbitrary labels. In addition, to prove a generalization result, we must assume some relation between features and labels. The separation margin gamma defined in our submission is one natural way to capture the feature-label relation, as discussed below. The details are given in Appendix E.
> 1. One interesting example is the noisy 2-XOR distribution introduced in the following paper: [Colin Wei, Jason D Lee, Qiang Liu, and Tengyu Ma. Regularization matters: Generalization and optimization of neural nets vs their induced kernel. arXiv preprint arXiv:1810.05369, 2018.]
> In the noisy 2-XOR distribution, the label is the XOR of the first two bits. Although there could be 2^d data examples, in Proposition E.1 we show that gamma is Omega(1/d). Interestingly, we further prove the following results for the noisy 2-XOR distribution.
> (a) In Proposition E.2, we prove that for any four points with the same last d-2 bits, if the width is less than sqrt{d-2}/4, then with probability 1/2, the finite-width NTK classifier fails on at least one point. This suggests that in the NTK regime, the width has to depend on at least 1/sqrt{gamma}. On the other hand, there is still a large gap between this lower bound and the 1/gamma^8 upper bound, and it is an interesting open question to close this gap.
> (b) In this paper, for a constant test accuracy, we prove a \tilde{O}(1/gamma^2) sample complexity upper bound for SGD and overparameterized networks. Such a sample complexity upper bound can also be shown for the infinite-width NTK, as discussed in Appendix E.2.2. As mentioned above, the margin gamma is Omega(1/d), which gives a sample complexity upper bound of \tilde{O}(d^2). On the other hand, [Wei, Lee, Liu, Ma] prove a sample complexity lower bound of d^2 for the infinite-width NTK. In other words, these bounds are tight up to logarithmic factors. We think it suggests that the notion of separation margin could be very useful, since the almost tight upper bound analysis highly relies on it.
> 2. A simpler example is the linearly separable case. In Appendix E.1, we show that if the data can be linearly separated with margin gamma, then Assumption 2.1 and 3.1 hold with margin at least gamma/2. In other words, when the data is linearly separable, the notion of margin in Assumption 2.1 and 3.1 could make a good use of this structure.
>
> We thank the reviewer for their comments. Most of them are handled in the revision, and we will keep working on them.

---

> > ### Comment · AnonReviewer1 · 2019-11-13
> > **Appendix E is Very Helpful**
> >
> > I would like to first reiterate that I would recommend accept for this paper regardless of the discussion on margin.
> >
> > Furthermore, I believe the addition of Appendix E giving more examples is quite helpful to understanding the \gamma parameter. In particular, I really like that section E.1 showed a connection with the classical notion of margin. And E.2 showed that the width is necessarily dependent on \gamma, so it's a natural quantity to study. Since \gamma can be seen as an important quantity on its own, I believe the "polylog dependence" in the title is now better justified.

---

### Official Review · AnonReviewer3 · 2019-10-23
**Official Blind Review #3**

**Rating:** 6

**Review:**

In this paper, the author shows that for a two-layer ReLU network,  it only requires a network width that is poly-logarithmic in the sample size n to get good optimization and generalization error bounds, which is better than prior results.

Overall, the paper is well written and easy to follow. However I still have some questions about this paper.

One of my major concerns is that there might be an important error in the proof of the main theorem. Specifically, in the proof of Theorem 2.2 (page 12), it says that due to lemma 2.5, $\hat R^{(t)}(\bar W)\leq \varepsilon$. However, Lemma 2.5 only shows that $|f(x_i,W_0,a)|$ is small, and the reason $\hat R^{(t)}(\bar W)$ can also be small is not explained in this paper at all. Based on Lemma 2.5, I can roughly get that $\hat R^{(0)}(\bar W) $ can be small, but the reason why $\hat R^{(0)}(\bar W) $ is small is unclear to me, especially when the network width m is only polylogarithmic in n and \varepsilon. Without a clear explanation on this issue, the theoretical results in this paper might be flawed, and the polylog claim might not be correct.

Moreover, this paper does not provide sufficient comparison with existing work. For example, Assumption 2.1 looks very similar to the assumption made in Cao & Gu (2019a). The definition of $\hat Q(W)$ has also been introduced in Cao & Gu (2019a). However these similarities are not mentioned in the paper at all. Moreover, the result of Lemma 2.6, which is also one of the selling points of this paper, is actually very similar to Fact D.4 and Claim D.6 in the following paper:
Allen-Zhu, Zeyuan, and Yuanzhi Li. "What Can ResNet Learn Efficiently, Going Beyond Kernels?." arXiv preprint arXiv:1905.10337 (2019).

Finally, the authors’ claim in the title that the width of the network is poly-logarithmic with the sample size n might be misleading. In fact, in Section 5, it has been discussed that in certain settings about the data distribution, $\frac 1\gamma$ is polynomial of n. However, the width is polynomial with $\frac 1\gamma$, which means the width is poly of n in these settings.

**Experience Assessment:**

I have read many papers in this area.

**Review Assessment: Checking Correctness Of Derivations And Theory:**

I carefully checked the derivations and theory.

**Review Assessment: Checking Correctness Of Experiments:**

I assessed the sensibility of the experiments.

**Review Assessment: Thoroughness In Paper Reading:**

I read the paper thoroughly.

---

> ### Author Response · Authors · 2019-11-11
> **Response to AnonReviewer #3**
>
> We thank the reviewer for their review.
>
> We apologize that the original proof of Theorem 2.2 was not clear enough. In fact, as AnonReviewer #2 points out, there was a term not handled in the original analysis. We have fixed this bug in the revision; however, the proof idea is still similar. For example, due to the exponential tail of the logistic loss, to show that \hat{R}^{(0)}(\bar{W}) is small, we only need to show that < \nabla f_i(W_0), \bar{W} > is large. Note that \bar{W} is defined as W_0+lambda\bar{U}, and < \nabla f_i(W_0), W_0 > is controlled by Lemma 2.5, and < \nabla f_i(W_0), \bar{U} > is concentrated around gamma by Lemma 2.3. Therefore with the chosen value of lambda, it holds that < \nabla f_i(W_0), \bar{W} > is large, and thus \hat{R}^{(0)}(\bar{W}) is small. To further handle \hat{R}^{(t)}(\bar{W}), we control < \nabla f_i(W_t) - \nabla f_i(W_0) , \bar{W} > using a standard NTK argument. More details of the proof of Theorem 2.2 are given at the end of Section 2 and in Appendix A. The required width still has only a polylogarithmic dependency on n, 1/delta, and 1/epsilon, but now it depends on 1/gamma^8.
>
> Regarding gamma, in Section 5 we give various estimates of gamma which depends on poly(n). However, these bounds are for arbitrary labels or random labels. In general it might be impossible to prove a polylogarithmic width for arbitrary or random labels, and this might also explain the poly(n) dependency in prior work, since their bounds usually hold for arbitrary labels. In addition, to prove a generalization result, we must assume some relation between features and labels. The separation margin gamma defined in our submission is one natural way to capture the feature-label relation, as discussed below. The details are given in Appendix E.
> 1. One interesting example is the noisy 2-XOR distribution introduced in the following paper: [Colin Wei, Jason D Lee, Qiang Liu, and Tengyu Ma. Regularization matters: Generalization and optimization of neural nets vs their induced kernel. arXiv preprint arXiv:1810.05369, 2018.]
> In the noisy 2-XOR distribution, the label is the XOR of the first two bits. Although there could be 2^d data examples, in Proposition E.1 we show that gamma is Omega(1/d). Interestingly, we further prove the following results for the noisy 2-XOR distribution.
> (a) In Proposition E.2, we prove that for any four points with the same last d-2 bits, if the width is less than sqrt{d-2}/4, then with probability 1/2, the finite-width NTK classifier fails on at least one point. This suggests that in the NTK regime, the width has to depend on at least 1/sqrt{gamma}. On the other hand, there is still a large gap between this lower bound and the 1/gamma^8 upper bound, and it is an interesting open question to close this gap.
> (b) In this paper, for a constant test accuracy, we prove a \tilde{O}(1/gamma^2) sample complexity upper bound for SGD and overparameterized networks. Such a sample complexity upper bound can also be shown for the infinite-width NTK, as discussed in Appendix E.2.2. As mentioned above, the margin gamma is Omega(1/d), which gives a sample complexity upper bound of \tilde{O}(d^2). On the other hand, [Wei, Lee, Liu, Ma] prove a sample complexity lower bound of d^2 for the infinite-width NTK. In other words, these bounds are tight up to logarithmic factors. We think it suggests that the notion of separation margin could be very useful, since the almost tight upper bound analysis highly relies on it.
> 2. A simpler example is the linearly separable case. In Appendix E.1, we show that if the data can be linearly separated with margin gamma, then Assumption 2.1 and 3.1 hold with margin at least gamma/2. In other words, when the data is linearly separable, the notion of margin in Assumption 2.1 and 3.1 could make a good use of this structure.
>
> We thank the reviewer for pointing out the relation to prior work. Here are the responses.
> 1. Assumption 2.1 and 3.1 are indeed similar to the assumption made in [Cao & Gu, 2019a]. The difference has been discussed in the related work Section of our original submission, and is now mentioned again below Assumption 3.1: [Cao & Gu, 2019a] assume separability in the RKHS induced by the NTK of the second layer, while we assume separability in the RKHS induced by the NTK of the first layer.
> 2. The quantity \hat{Q} is indeed analyzed in [Cao & Gu, 2019a], and also [Nitanda & Suzuki]. We discuss it at the beginning of Section 2.2 in the revision.
> 3. Lemma 2.6 is indeed similar to Fact D.4 and (seemingly) Claim D.5 of [Allen-Zhu & Li, 2019], where the squared loss is considered. We discuss it below Lemma 2.6 in the revision. On the other hand, we still want to highlight Lemma 2.6 since it plays an important role in proving a polylog(1/epsilon) width (see the discussion at the end of Section 2, bullet 2).
>
> We would very much like to discuss any further questions!

---

> > ### Comment · AnonReviewer2 · 2019-11-13
> > **Corrected Typos**
> >
> > [Zeyuan & Li, 2019] should be [Allen-Zhu & Li, 2019]

---

> > > ### Author Response · Authors · 2019-11-14
> > > **The typo has been corrected in the response**
> > >
> > > We thank AnonReviewer2  for the correction; we apologize for the oversight.  We have corrected our response, and confirm that the reference is correct in our revised submission.

---

> ### Comment · AnonReviewer2 · 2019-11-13
> **Is that possible to reconsider your decision for the paper?**
>
> I think authors already addressed the comments very carefully.

---

> ### Comment · AnonReviewer3 · 2019-11-14
> **Official Blind Review #3**
>
> I think the revision have fixed the problem I mentioned and I will increase the score.

---

### Official Review · AnonReviewer2 · 2019-10-27
**Official Blind Review #2**

**Rating:** 8

**Review:**

Summary :
1. Classification task with shallow Relu and logistic loss.
2. Showing fast global convergence rate with polylog width for both training and generalization error under appropriate assumptions

Overall, this paper is very exciting and surprising. At some point, I was trying to prove such results, but couldn’t get it. This paper should be accepted as an oral presentation in ICLR. If the authors can address some of the questions in the comments, I will be happy to increase the score.

Advantages:
1. Better results for classification task with shallow Relu in terms of global convergence rate and network width
2. Showing the essence of the power of over-parameterization that the weights don’t change much
3. Clear logic and proof
4. Also discuss the stochastic GD/generalization error? (I didn’t read that part)



Disadvantages:
1.Bug in proving Theorem 2.2, a larger lambda is needed (but won’t influence the polylog result). See the below for a fix.
2.In the proving sketch, why ||W_t-W_0||_F=O(ln t)? In the proof, it looks like ||W_t-W_0||^2_F=O(t), as in the third equation on page 12. More explanation about proof sketch is needed.
3.Typo
a.The \odot operation in Equation 5.1 is not defined. According to later computation, this operation seems to be the hadamard product between two vectors. But this notation is not widely used and some brief introduction will be benefinitial.
b.On page 1, “… and standard Rademacher tools but exploiting how little the….”, “but” seems to be a typo.
c.On page 2, “also suffices via a smoothness-based generalization bound”, “suffices” should be “suffice”.
d.Last formula in page 11 missing a “>0” in the indicator function.
4.(optional) why using ||W_t-W_0||_F instead of ||w_r(t)-w_r(0)||_2, which used in previous work for square loss, for the analysis? Is there any benefit or restriction here?
5.(optional) Give more insights about intermediate quantities such as \hat R^(t), \bar{W}, etc.

Comments:
1. More arguments for polylog width in last section needed. E.g., give a specific case where the gamma in Assumption 2.1 is constant, or comparable to the smallest eigenvalue of NTK; otherwise in the worst case, gamma can be as bad as  the smallest eigenvalue of NTK over n, which ruins the polylog results. To be more specific, we can always set q to be the uniform distribution over [n], then \|q\odot y\|_2 is indeed 1/\sqrt{n}, hence \gamma_1\leq \sqrt{\lambda_{max}(K_1)/n}. If K_1 has constant spectral norm(which is the case if all the data points are orthogonal to each other), then \gamma_1 will depend on 1/n.
2. For the over-parameterization theory, more references are needed. https://arxiv.org/abs/1902.01028 [Allen-Zhu, Li] is about generalization bound for the over-parametrized networks, https://arxiv.org/abs/1810.12065 [Allen-Zhu, Li, Song] and https://arxiv.org/abs/1905.10337 [Allen-Zhu, Li] are about the over-parameterization bound for more than two-layer networks. https://arxiv.org/abs/1906.03593 [Song, Yang] obtains a better width bound for two-layer neural networks under the framework of https://arxiv.org/abs/1810.02054 [Du, Zhai, Poczos, Singh].
3. Theorem 2.2 shows that the average loss converges. Does this imply after training for T steps, we obtain good weights with small logistic loss? Can you get results showing the loss is decaying, like Theorem 4.1 in https://arxiv.org/abs/1810.02054 [Du, Zhai, Poczos, Singh]?
4. On page 8, the lower bound of \lambda_0 is given as \delta/n^2. Is this bound tight? Is this lower bound achievable?
5. Under what assumptions can we prove o(log n), say poly(log log n) width?
6. What is the role of logistic loss in the proof? In general, if we replace logistic loss with square loss, will this make it harder to train neural networks?


The original analysis might has some flaw/bug:

In the proof of Theorem 2.2, top of page 12, to show \hat R^{(t)}(\bar W)<= \epsilon/4, the term y_i <\nabla f_i (W_t) - \nabla f_i (W_0) , W_0> seems to be forgotten to consider.

This could be a fix.

Note that above term equals y_i/m \sum_{r=1}^m ( 1_{[< w_{r, t}, x_i> >= 0]} - 1_{[< w_{r, 0}, x_i> >= 0]} ) < w_{r, 0}, x_i >. We can use concentration to bound <w_{r, 0}, x_i>, such that with high probability, it will be no larger than polylog(n). Correspondingly, we know this term won’t be too small. Adding this extra polylog factor into lambda, we can fix the proof.


**Experience Assessment:**

I have published in this field for several years.

**Review Assessment: Checking Correctness Of Derivations And Theory:**

I carefully checked the derivations and theory.

**Review Assessment: Checking Correctness Of Experiments:**

N/A

**Review Assessment: Thoroughness In Paper Reading:**

I read the paper thoroughly.

---

> ### Author Response · Authors · 2019-11-11
> **Response to AnonReviewer #2**
>
> We thank the reviewer for their review and support.
>
> We are particularly grateful to the reviewer for catching the missing y_i <\nabla f_i (W_t) - \nabla f_i (W_0) , W_0> term in our original analysis. We have fixed this bug in the revision, using the quantity ||w_r(t)-w_r(0)||_2 suggested by the reviewer. In the suggested fix, there is a 1/m factor which should actually be 1/sqrt{m}; in our current proof, the additional 1/sqrt{m} factor we need comes from ||w_r(t)-w_r(0)||_2. Other parts of the proof are the same as before. The required width still has only a polylogarithmic dependency on n, 1/delta, and 1/epsilon, but now it depends on 1/gamma^8.
>
> We have also included in Appendix E some concrete examples where the gamma in Assumption 2.1 and 3.1 is large.
> 1. One interesting example is the noisy 2-XOR distribution introduced in the following paper: [Colin Wei, Jason D Lee, Qiang Liu, and Tengyu Ma. Regularization matters: Generalization and optimization of neural nets vs their induced kernel. arXiv preprint arXiv:1810.05369, 2018.]
> In the noisy 2-XOR distribution, the label is the XOR of the first two bits. Although there could be 2^d data examples, in Proposition E.1 we show that gamma is Omega(1/d). Interestingly, we further prove the following results for the noisy 2-XOR distribution.
> (a) In Proposition E.2, we prove that for any four points with the same last d-2 bits, if the width is less than sqrt{d-2}/4, then with probability 1/2, the finite-width NTK classifier fails on at least one point. This suggests that in the NTK regime, the width has to depend on at least 1/sqrt{gamma}. On the other hand, there is still a large gap between this lower bound and the 1/gamma^8 upper bound, and it is an interesting open question to close this gap.
> (b) In this paper, for a constant test accuracy, we prove a \tilde{O}(1/gamma^2) sample complexity upper bound for SGD and overparameterized networks. Such a sample complexity upper bound can also be shown for the infinite-width NTK, as discussed in Appendix E.2.2. As mentioned above, the margin gamma is Omega(1/d), which gives a sample complexity upper bound of \tilde{O}(d^2). On the other hand, [Wei, Lee, Liu, Ma] prove a sample complexity lower bound of d^2 for the infinite-width NTK. In other words, these bounds are tight up to logarithmic factors. We think it suggests that the notion of separation margin could be very useful, since the almost tight upper bound analysis highly relies on it.
> 2. A simpler example is the linearly separable case. In Appendix E.1, we show that if the data can be linearly separated with margin gamma, then Assumption 2.1 and 3.1 hold with margin at least gamma/2. In other words, when the data is linearly separable, the notion of margin in Assumption 2.1 and 3.1 could make a good use of this structure.
>
> Here are responses to the disadvantages pointed out by the reviewer:
> 1. Discussed above.
> 2. We can show that ||W_t-W_0||_F=O(lambda), where lambda is defined in Theorem 2.2 and depends on ln(1/epsilon). On the other hand, to get an error of epsilon, we need roughly 1/epsilon steps. Therefore we said that ||W_t-W_0||_F=O(ln t), which is actually an incomplete argument since it only considers epsilon. What we want to highlight is that in our setting, to make the width depend only on ln(1/epsilon), it is important to have an upper bound on the GD movement which also only depends on ln(1/epsilon). More discussion is given at the end of Section 2 (bullet 2).
> 3. The typos have been fixed.
> 4. The current proof uses both ||W_t-W_0||_F and ||w_r(t)-w_r(0)||_2. The quantity ||W_t-W_0||_F is important in Lemma 2.6, and we are not sure if we can avoid using it completely.
> 5. As discussed at the beginning of Section 2 of the revision, Assumption 2.1 and 3.1 are basically separability assumptions in the RKHS induced by the NTK. The training feature is mapped to \phi_i which lies in the RKHS, while \bar{v} given by Assumption 2.1 and 3.1 is a separator in the RKHS. Our analysis then basically deals with finite-width samples of these points in the RKHS, e.g., \nabla f_i(W_0) consists of samples of \phi_i, while \bar{W} is constructed from samples of \bar{v}.

---

> > ### Author Response · Authors · 2019-11-11
> > **Response to AnonReviewer #2 Cont.**
> >
> > Here are responses to the reviewer's comments:
> > 1. Discussed above.
> > 2. We thank the reviewer for pointing out the references and have included them.
> > 3. We think it is possible to show a decreasing risk using the smoothness of the logistic loss and the NTK analysis, but have not finished it. We will keep working on it.
> > 4. We do not know whether this lower bound is tight or not.
> > 5. It seems unlikely to prove an o(log n) width with the current proof techniques. In fact, a polylog(n) width is already required to ensure the finite-width NTK has a positive margin (cf. Lemma 2.3).
> > 6. One key step in our analysis is to show the representation result that \hat{R}^{(t)}(\bar{W}) is small. To show such a result with a polylog(n, 1/delta, 1/epsilon) width, we need some assumption on the relation between features and labels. For classification, the margin naturally captures the feature-label relation, and it works well with the logistic loss. If we want to prove a similar result for the squared loss, then we should need a similar assumption. In addition, the logistic loss and its derivative (the sigmoid function) allows a clean generalization analysis.

---

### Author Response · Authors · 2019-11-11
**List of changes made in the revision**

1. As pointed out by AnonReviewer #2, there was a missing term in the original proof of Theorem 2.2; this oversight has now been fixed. In more detail, the missing term is small for reasons common to NTK analyses, namely that weights stay close to initialization; the new proof is sketched at the end of Section 2, and appears in full in Appendix A.  This proof adjustment caused the dependency on 1/gamma in the network width to worsen to at most 1/gamma^8 in Theorems 2.2, 3.2, 4.1 and Corollary 3.3; the other polylogarithmic terms are unchanged.
2. All reviewers requested further discussion of the margin parameter gamma.  With the aim of demonstrating that gamma is both a natural quantity, and that it is large in interesting cases (notably, not breaking the "polylogarithmic" promise), the revision contains a new Appendix E providing a variety of examples.  A first example is the linear case, where gamma is the usual linear separation margin.  A more interesting example is the "noisy 2-XOR" problem, which has the xor on 2 bits in d dimensions, and for which we can show gamma is indeed a natural parameter: (a) a sample complexity lower bound of d^2 (proved in prior work), (b) a margin lower bound 1/d, (c) a sample complexity upper bound in the SGD case of d^2, thus showing the margin-based analysis is tight in this case, (d) a width lower bound 1/sqrt{gamma} below which the problem becomes nonseparable by the NTK with constant probability.
3. In the abstract and Section 1, some typos have been fixed, and references have been added.
4. At the beginning of Section 2, explanations of the separability assumption have been added. At the beginning of Section 2.2, more discussion on the quantity \hat{Q} has been added. The proof of Lemma 2.6 has been moved to Appendix A, while some discussion has been added below Lemma 2.6.
5. In Section 3, some discussion on separability assumptions made in prior work has been added.
6. In Section 5, more explanations have been added. The upper bound on the margin for random labels is now formally stated in Proposition 5.2. Some typos have been fixed.

---

> ### Comment · AnonReviewer2 · 2019-11-13
> **Good revisions**
>
> I like to increase my score to 10, but it seems the maximum allowed is only 8. I still think this paper should be at least an oral presentation of ICLR. I hope Reviewer 3 can reconsider his opinion.

---

> > ### Comment · AnonReviewer1 · 2019-11-13
> > **Agreed with Reviewer 2**
> >
> > I would like to also urge Reviewer 3 to reconsider their opinion. I believe the authors have adequately addressed all of the concerns we have raised, and this result is quite tight compared to existing results.

---

> > > ### Author Response · Authors · 2019-11-14
> > > **Thank you for the support!**
> > >
> > > If there are any additional comments on the revision, we would very much like to hear them and improve our submission correspondingly.

---

### Decision · Program_Chairs · 2019-12-19

**Decision:**

Accept (Poster)

**Comment:**

This paper studies how much overparameterization is required to achieve zero training error via gradient descent in one hidden layer neural nets. In particular the paper studies the effect of margin in data on the required amount of overparameterization. While the paper does not improve in the worse case in the presence of margin the paper shows that sometimes even logarithmic width is sufficient. The reviewers all seem to agree that this is a nice paper but had a few mostly technical concerns. These concerns were sufficiently addressed in the response. Based on my own reading I also find the paper to be interesting, well written with clever proofs. So I recommend acceptance. I would like to make a suggestion that the authors do clarify in the abstract intro that this improvement can not be achieved in the worst case as a shallow reading of the manuscript may cause some confusion (that logarithmic width suffices in general).